# Innovative Materials for Energy Storage and Conversion

**DOI:** 10.3390/molecules27133989

**Published:** 2022-06-21

**Authors:** Shi Li, Shi Luo, Liya Rong, Linqing Wang, Ziyang Xi, Yong Liu, Yuheng Zhou, Zhongmin Wan, Xiangzhong Kong

**Affiliations:** 1College of Mechanical Engineering, Hunan Institute of Science and Technology, Yueyang 414006, China; 11999429@hnist.edu.cn (S.L.); ls19971230@163.com (S.L.); 822011110434@vip.hnist.edu.cn (L.W.); 812111110115@vip.hnist.edu.cn (Z.X.); 822111110443@vip.hnist.edu.cn (Y.L.); 822111110444@vip.hnist.edu.cn (Y.Z.); zhongminwan@hotmail.com (Z.W.); 2Institute of New Energy, Hunan Institute of Science and Technology, Yueyang 414006, China

**Keywords:** metal chalcogenides, layered and non-layered structures, modification strategies, reaction mechanism, sodium-ion batteries

## Abstract

The metal chalcogenides (MCs) for sodium-ion batteries (SIBs) have gained increasing attention owing to their low cost and high theoretical capacity. However, the poor electrochemical stability and slow kinetic behaviors hinder its practical application as anodes for SIBs. Hence, various strategies have been used to solve the above problems, such as dimensions reduction, composition formation, doping functionalization, morphology control, coating encapsulation, electrolyte modification, etc. In this work, the recent progress of MCs as electrodes for SIBs has been comprehensively reviewed. Moreover, the summarization of metal chalcogenides contains the synthesis methods, modification strategies and corresponding basic reaction mechanisms of MCs with layered and non-layered structures. Finally, the challenges, potential solutions and future prospects of metal chalcogenides as SIBs anode materials are also proposed.

## 1. Introduction

Based on the deterioration of the global environment and the gradual exhaustion of fossil fuels, the exploration of clean energy sources has been accelerating, and suitable energy storage equipment has also been extensively studied [1,2,3,4,5]. In recent years, lithium-ion batteries (LIBs) have been extensively applied for portable electronic devices and electric vehicles. However, limited lithium resources limit the development of lithium-ion batteries [6,7,8,9]. Recently, sodium ion batteries are promising as a substitute to lithium ion batteries due to the natural abundance of sodium element and the similar electrochemical reaction mechanisms as the LIBs [10,11,12]. The numerous materials including carbonaceous materials [13,14,15], metal oxides [16,17], metal phosphides [18,19], and metal chalcogenides (MCs) have been studied as anodes for SIBs. Carbonaceous materials exhibit excellent cycle stability but suffer from low reversible capacity. Although the metal oxides have high theoretical capacity, there are severe problems of volume expansion and rapid capacity fade during the reaction. Metal phosphides exhibits a high theoretical capacity, but dangerous toxicity limits its practical application [9]. Compared with the above materials, metal chalcogenides possess an overwhelming capacity advantage, excellent cycle stability, low redox potential, higher conductivity and good controllability [20]. The interaction between the guest ion of the layered structure and the chalcogenide lattice is weak, which promotes the migration of ions and effectively reduces the energy barrier required to initiate the intercalation reaction. Therefore, metal chalcogenides have received extensive attention and are considered as ideal anode electrode materials for sodium-ion batteries.

To date, metal sulfides and metal selenides have been extensively studied. For example, in 1980, Newman et al. reported that TiS_2_ was used as the electrode material for SIBs, and then, metal chalcogenides were widely used in electrode materials for sodium-ion batteries [21]. Zhang et al. reported FeSe_2_ microspheres as an anode material for sodium-ion batteries with an excellent high-rate performance of 388 mA h g^−1^ at 10 C [22]. Chen et al. synthesized the FeS_2_/CNT neural network nanostructure composite material (FeS_2_/CNT-NN) as anode material for a sodium-ion battery [23]. After 1800 cycles at a current of 1 A g^−1^, the electrode maintained a high reversible capacity of 309 mA h g^−1^. Even when the current density is increased to 22 A g^−1^, the reversible capacity remains at 254 mA h g^−1^ after 8400 cycles. Liu et al. synthesized the heterojunction SnSe_2_/ZnSe nanobox coated by a polydopamine shell (SnSe_2_/ZnSe@PDA) for the anode of SIBs, which has a superior reversible capacity of 744 mA h g^−1^ at 100 mA g^−1^ after 200 cycles [24]. The heterostructure and hollow structure effectively improve the thermodynamic stability, promote the reaction kinetics, and alleviate the volume expansion, improving the electrochemical performance. Huang et al. reported the heterostructured core–shell Bi_2_S_3_@Co_9_S_8_ complex hollow particles (CHPs) as anode materials for SIBs, which exhibit high reversible capacities of 461 mA h g^−1^ at 0.1 A g^−1^ after 100 cycles [25]. Bi_2_S_3_@Co_9_S_8_ CHPs has rich redox properties and a special hollow structure; as a result, when used as a negative electrode material, it shows excellent electrochemical performance. To further improve MCs’ electrochemical properties and facilitate their practical applications, further optimization of mechanical stability and kinetic properties is required.

Herein, plenty of MCs (M = Fe, Co, Mo, V, W, Ni, S, Sb, Zn, Cu, Mn, Ti, Bi or mixture of them) have been widely reported and used as anode materials for SIBs [21]. What is more, many methods to solve the severe volume expansion and slow kinetics of metal chalcogenides in the reaction process have been reported. In this review, we summarize the current advances of metal chalcogenides for SIBs. This work reviews the research progress of MCs as advanced materials for high-performance SIBs in recent years. The synthesis method, structure composition, reaction mechanism, in situ characterization and performance modification methods are discussed. Moreover, based on the structure and conductivity of metal chalcogenides, we summarized several strategies to suppress the drawbacks of MCs as anode materials for SIBs, such as designing nanostructure with various morphologies, carbon coating, hierarchical structure and hollow porous structure. Finally, a brief discussion on the future development and perspectives of metal chalcogenides materials as advanced anodes for SIBs has been made. Compared with the previously reported literature, we have elucidated and summarized the detailed reaction mechanisms of numerous MCs during sodiation/desodiation processes based on advanced in situ/ex situ characterization techniques, which provides significant theoretical guidance for developing MCs with enhanced sodium storage properties.

## 2. Metal Chalcogenides for SIBs

Metal chalcogenides can generally be divided into two parts: non-layered material and layered structure material. The non-layered substances are also very popular due to its low price and high theoretical capacity advantages. non-layered substances crystallize in three dimensions through atoms or chemical bonds, reflecting the non-layered nature of their bulk crystals. Layer-structured substances have been the focus of more attention due to the unique layered structure and excellent electrochemical properties. For layered materials, the strong chemical bonds in each layer connect the in-plane atoms to each other, and these layers are stacked together through weak Van der Waals force interactions to form bulk crystals. In addition, a weaker interaction between the guest ions and chalcogenide lattice for layered structures enables faster ion migration, lowering the energy barrier required to initiate the intercalation reaction. At present, numerous methods, such as solvothermal reaction, spray pyrolysis, chemical vapor deposition (CVD), electrospinning, exfoliation, sulfation/selenization, reflux, ball milling, etc. have been utilized for the synthesis of nanostructured MCs materials.

### 2.1. Non-Layer Structured MCs

#### 2.1.1. Fe-Based Chalcogenides

##### Iron Sulfide

As a common by-product of coal production, iron disulfide with advantages of low price and high theoretical capacity has received significant attention in recent years. However, there is a lack of consensus on the charge storage mechanism of FeS_2_. Na_x_FeS_2_ was postulated to be the intermediate when x < 2.0, which was accompanied by a valence change of S from (S–S)^2−^ to S^2−^. Upon x > 2.0, the intercalated species Na_2_FeS_2_ is fully converted to metallic Fe and Na_2_S. However, there is a lack of consensus on the charge storage mechanism of FeS_2_; the most important of which is whether Na_2_S and Fe can be formed above 0.8 V. Chen et al. synthesized FeS_2_@graphene@carbon nanofibers (FeS_2_@G@CNF) through electrospinning [18]. In situ XRD was used to track the iron disulfide in the sodiation/desodiation process, as shown in Figure 1a. The FeS_2_@G@CNF also exhibits excellent cycling stability (400 mA h g^−1^ at 200 mA g^−1^ after 370 cycles) (Figure 1b). The results confirmed the following Na^+^ storage mechanism (Equations (1) and (2)):(1)$${\mathrm{FeS}}_{2}\;+\;\mathrm{xNa}+{\mathrm{xe}}^{-}\to {\mathrm{Na}}_{x}{\mathrm{FeS}}_{2}\;\left(0\;<\;x\;<\;2\right)$$
(2)$${\mathrm{Na}}_{x}{\mathrm{FeS}}_{2}\;+\;\left(4\;-\;x\right){\mathrm{Na}}^{+}\;{+\;(4\;-\;x)\;e}^{-}\to {\mathrm{Fe}\;+\;2\mathrm{Na}}_{2}S$$

Various nanostructured FeS_2_ forms are widely used in sodium ion batteries. For example, as shown in Figure 1c,d, Zhang et al. synthesized a neural network nanostructured composite material (FeS_2_/CNT-NN) by a one-pot solvothermal method. The unique neural network structure provides high surface area and small FeS_2_ particle size for sufficient sodiation and desodiation, and it provides sufficient space and mechanical integrity for volume expansion. Thus, the FeS_2_/CNT-NN showed superior rate and cycling performance with capacities of 309 and 254 mA h g^−1^ retained after 1800 and 8400 cycles at 1 and even 22 A g^−1^, respectively [23]. As shown in Figure 1e,f, Zhao et al. report a novel FeS_2_@C hybrid multilayer structure, which is composed of ultra-thin nanoflakes building blocks and also exhibits good electrochemical performance [26]. The multi-level structure has good availability in constructing a material’s structure. Zhang et al. synthesized the composite FeS_2_@CF-NS by the electrospinning method, in which FeS_2_ nanoparticles were encapsulated in S and N co-doped carbon layers, and FeS_2_ nanosheets were attached to the surface (Figure 1g). The FeS_2_@CF-NS yielded a capacity of 637.1 mA h g^−1^ after 400 cycles at 1 A g^−1^ and an excellent rate performance of 431.1 mA h g^−1^ at 5 A g^−1^, as shown in Figure 1h,i [27]. The N and S-co-doped carbon layer alleviates the volume expansion during the reaction, and it also adsorbs more Na^+^ for electrochemical reaction. In addition, a large number of defect sites generated by the high-temperature carbonization of carbon fiber are beneficial to maintain the high rate and stable cycle performance.

**Figure 1 molecules-27-03989-f001:**
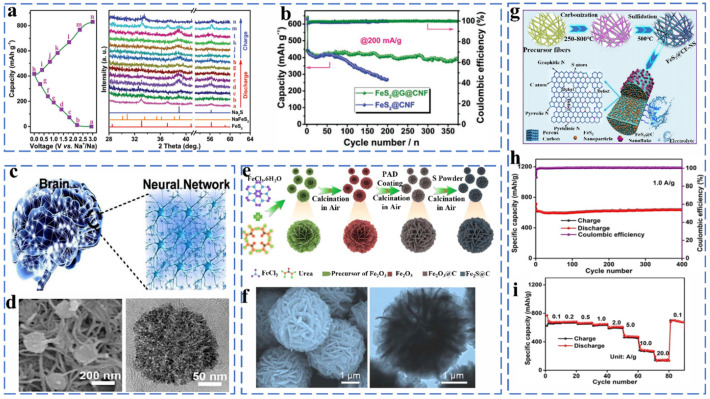
(**a**) The ex situ XRD patterns of FeS_2_@G@CNF as Na-ion battery anodes during the discharge and charge process, (**b**) discharge capacities at 200 mA g^−1^. Reprinted with permission from Ref. [18]. Copyright 2019, copyright Chen, C. (**c**) Schematic illustration of the FeS_2_/CNT neural network nanostructure composite (FeS_2_/CNT-NN), (**d**) SEM and TEM images. Reprinted with permission from Ref. [28]. Copyright 2018, copyright Chen, Y. (**e**) Representative illustration of the synthetic process of multi-layer architecture of FeS_2_@C hybrids, (**f**) SEM and TEM images. Reprinted with permission from Ref. [29]. Copyright 2019, copyright Wu, X. (**g**) Schematic illustration of the synthesis process of the FeS_2_@CF-NS, (**h**) cycling performance at current density of 1.0 A g^−1^, (**i**) Rate performance at different current densities. Reprinted with permission from Ref. [30]. Copyright 2021, copyright Shao, L.

Shao et al. synthesized an urchin-like FeS_2_ layered structure through a hydrothermal approach, which is wrapped with reduced graphene oxide and N-doped multi-walled carbon nanometers (FeS_2_@NCNTs@rGO) [31]. The FeS_2_@NCNTs@rGO exhibits splendid electrochemical performance, including a high capacity of 558 mA h g^−1^ at 0.1 A g^−1^ and superior rate capacity of 419 mA h g^−1^ at 5 A g^−1^. The layered structure interconnected with rGO and CNTs is conducive to the infiltration of electrolyte and alleviates the volume expansion caused by the sodiation/desodiation reaction. Ding et al. successfully synthesized spindle-like double-component composites constructed with an FeS_2_ core and nitrogen/sulfur co-doped carbon shell (FeS_2_@NSC), which exhibits a high specific rate capacity of 536.5 mA h g^−1^ at 1 A g^−1^ [32].

##### Iron Selenide

Metal selenides have received extensive attention as anode materials for SIBs because of their much higher initial Coulombic efficiency than oxides, more stable cycle performance than sulfide and alloys, and better structural stability and lower toxicity than phosphides. However, selenide still faces the challenge of low conductivity and large volume expansion. Strategies for constructing nanostructured materials have been well applied to selenide, such as self-assembled nanostructures, nanostructured composite materials, mesoporous materials, and layered 3D conductive networks. Different methods of synthesizing metal selenides have been introduced such as the hydrothermal method, solvothermal method synthesis, pulsed laser deposition, electrodeposition method, chemical bath deposition, chemical vapor deposition and template-directed method. Zhang et al. prepared FeSe_2_ microspheres composed of numerous nano-octahedrons by a simple hydrothermal method. The FeSe_2_ microspheres deliver the capacity 372 mA h g^−1^ after 2000 cycles [33]. Pan et al. synthesized a Na-ion battery anode material (FeSe_2_@NC) with a hollow structure and high electrical conductivity, and they investigated its sodium storage mechanism by in situ XRD (Figure 2a) [34]. FeSe_2_@NC shows good electrochemical properties (401.3 mA h g^−1^ after 2000 cycles at 5.0 A g^−1^) as sodium-ion anode because of the design of the hollow structure and the doping of N atoms. When the FeSe_2_ is used as the anode material of SIBs, during the discharging process, FeSe_2_ will appear as an intermediate product Na_x_FeSe_2_. Finally, Na_2_Se is generated in a fully discharged state, and then, the final product is FeSe_2_ in a charged state. This shows that this reaction is reversible. The reaction of FeSe_2_ during the cycle can be summarized as follows (Equations (3)–(7)):

Discharged processes
(3)$${\mathrm{FeSe}}_{2}{\;+\;\mathrm{xNa}}^{+}\;{+\;\mathrm{xe}}^{-}\to {\mathrm{Na}}_{x}{\mathrm{FeSe}}_{2}$$
(4)$${\mathrm{Na}}_{x}\mathrm{FeSe}\;+\;\left(2\;-\;x\right){\mathrm{Na}}^{+}{\;+\;(2\;-\;x)e}^{-}\to {\mathrm{FeSe}\;+\;\mathrm{Na}}_{2}\mathrm{Se}$$
(5)$${\mathrm{FeSe}}_{2}{\;+\;2\mathrm{Na}}^{+}\;{+\;2e}^{-}\to {\mathrm{Fe}\;+\;\mathrm{Na}}_{2}\mathrm{Se}$$

Charge processes
(6)$${\mathrm{Fe}+2\mathrm{Na}}_{2}\mathrm{Se}\to {\mathrm{Na}}_{x}{\mathrm{FeSe}}_{2}+\left(4\;-\;x\right){\mathrm{Na}}^{+}{\;+\;(4\;-\;x)e}^{-}$$
(7)$${\mathrm{Na}}_{x}{\mathrm{FeSe}}_{2}\to {\mathrm{FeSe}}_{2}{\;+\;\mathrm{xNa}}^{+}{\;+\;\mathrm{xe}}^{-}$$

However, FeSe_2_ have poor cycling performance and low conductivity during the cycle. The method of coupling with carbon-containing materials can effectively alleviate the volume expansion of the electrode and increase the conductivity. As shown in Figure 2b, Muhammad et al. used a simple wet-chemistry method to embed core–shell (FeSe_2_/C) nanostructures into a carbon nanotube (CNT) frame as a sodium-ion anode material. The CNT frame constructs a three-dimensional conductive network, and the macropores provide enough space for the high-quality load of FeSe_2_/C. The CNT/FeSe_2_/C exhibited a capacity 395 mA h g^−1^ even after 200 cycles at 500 mA g^−1^ [35]. Fan et al. successfully prepared 3D hierarchical hollow nanocubes constructed from one-dimensional FeSe_2_@C core–shell nanorods through the thermally induced selenization process of the Prussian blue microcube precursor (Figure 2c,d). FeSe_2_@C delivered a discharge capacity of 212 mA h g^−1^ after 3000 cycles at 10 A g^−1^ [36]. Lv et al. report the synthesis of a composite of ultra-fine FeSe NPs and carbon nanofiber aerogel (CNFA); FeSe was uniformly embedded in interconnected three-dimensional (3D) carbon nanofibers, which improved the electronic conductivity and structural stability [37]. Furthermore, density functional theory (DFT) is calculated to investigate the Na adsorption process on the FeSe (001) surface and at the FeSe/carbon layer interface at atomic levels the interaction and electron transport of an FeSe-CNFA-T hybrid. The FeSe/carbon layer interface has the lowest activation barrier (0.14 eV), which is followed by the FeSe (001) surface (0.34 eV) and carbon layer (0.42 eV). The figure shows the minimum energy path of sodium-ions inserted along the axis. The small energy barrier at the interface of the FeSe/carbon layer can improve the conductivity of sodium ion. The adsorption energy is defined as Eb = (Ena/surf − Esurf)/n − ENa, where ENa/surf and Esurf are the total energies of the surface with and without n Na atoms adsorbed, respectively, and ENa is the energy of one Na atom in the bulk position. A more negative Eb indicates a more stable adsorption. For the most stable adsorption site for a Na atom between two Se atoms on the surface, the binding energies at the interface of the carbon layer, FeSe and FeSe/carbon layer are 0.30, −1.82 and −1.95 eV, respectively. The addition of the outer carbon layer makes the charge configuration of Na atoms transfer from the outer periphery of Fe and Se atoms to the carbon layer, causing the electron density around Na atoms to rearrange. These results indicate that Fe has a good sodium ion storage performance. The FeSe-CNFA-700 sample exhibited a high rate capacity of 291 mA h g^−1^ at 5 A g^−1^. FeSe-CNFA-700 shows excellent electrochemical performance, which is due to the fact that FeSe NPs can shorten the ion transmission path and improve the reaction kinetics. The three-dimensional carbon nanofiber aerogel effectively improves the conductivity and reduces the pulverization rate. Xiong et al. synthesized a composite material with micro-flower-like two-dimensional core/shell FeSe/carbon nanosheets. The specific capacity remains at 214.6 mA h g^−1^ at 1 A g^−1^ after 140 cycles [38].

#### 2.1.2. Co-Based Chalcogenides

##### Cobalt Sulfide

Cobalt sulfide exists in various stoichiometric forms, including CoS, CoS_2_, Co_3_S_4_, Co_9_S_8_, CoS_x_, CoS_1−x_, etc. CoS_2_ has been widely studied because of its high theoretical capacity (872 mA h g^−1^). Studies have shown that CoS_2_ is similar to FeS_2_ with a Pa3^−^ space group and has a large potential value in the field of sodium ion battery anodes. Zhao et al. successfully synthesized CoS_2_@C composites with hollow nanocubic structures by simple sulfidation and carbon capping, and they investigated the sodium storage mechanism of the composites by in situ XRD (Figure 3a,b). From the images, it can be found that when the electrode is discharged from OCV to 1.6 V, the diffraction peak of CoS_2_ is significantly weakened, and the corresponding diffraction peak of Na_2_S is increasing. This indicates that the conversion reaction of CoS_2_ has occurred. With further discharge, the CoS_2_ phase gradually disappeared, and the Na_2_S peak reached the strongest. During the charging process, the peaks of metallic Co disappeared, while the peaks of CoS_2_ reappeared. It shows that the CoS_2_ phase has a good reversible conversion reaction [39]. The reaction mechanism of CoS_2_ is as follows (Equations (8)–(12)):

Sodiation:(8)$${\mathrm{CoSe}}_{2}{\;+\;\mathrm{xNa}}^{+}{\;+\;\mathrm{xe}}^{-}\to {\mathrm{Na}}_{x}{\mathrm{CoSe}}_{2}$$
(9)$${\mathrm{Na}}_{x}{\mathrm{CoSe}}_{2}{\;+\;\mathrm{xNa}}^{+}{\;+\;\mathrm{xe}}^{-}\to {\mathrm{CoSe}\;+\;\mathrm{Na}}_{2}\mathrm{Se}$$
(10)$${\mathrm{CoSe}\;+\;2\mathrm{Na}}^{+}{\;+\;2e}^{-}\to {{\mathrm{Co}}^{+}\;+\;\mathrm{Na}}_{2}\mathrm{Se}$$

Desodiation:(11)$${\mathrm{Co}\;+\;2\mathrm{Na}}_{2}\mathrm{Se}\to {\mathrm{Na}}_{x}{\mathrm{CoSe}}_{2}\;+\;\left(4\;-\;x\right){\mathrm{Na}}^{+}{\;+\;(4\;-\;x)e}^{-}$$
(12)$${\mathrm{Na}}_{x}{\mathrm{CoSe}}_{2}\to {\mathrm{CoSe}}_{2}{\;+\;\mathrm{xNa}}^{+}{\;+\;\mathrm{xe}}^{-}$$

Zhang et al. successfully prepared AGC-CoS_2_@NCNFs composites by anchoring graphitic carbon-coated CoS_2_ nanoparticles on one-dimensional N-doped porous carbon nanofibers using an electrostatic spinning strategy (Figure 3c). Attributed to the one-dimensional porous carbon nanofiber structure and the amorphous and graphitic carbon coatings, the CoS_2_@NCNFs exhibited excellent electrochemical performance as the anode of sodium-ion batteries. A large specific capacity of 876 mA h g^−1^ was exhibited after 200 cycles at 100 mA g^−1^, and a multiplicative capacity of 148 mA h g^−1^ was also provided at 3.2 A g^−1^ (Figure 3d) [40]. As shown in Figure 3e, Zhao et al. reported metal–organic framework-derived Co_9_S_8_@carbon yolk shell nanocages (Co_9_S_8_@CYSNs). Co_9_S_8_@CYSNs have a unique hierarchical structure and uniformly dispersed Co_9_S_8_ nanoparticles, which shorten the Na^+^ diffusion distance and provide a fast electron transport channel. The material showed superior cycling performance with a capacity of 461 after 400 cycles at 1.0 A g^−1^ and 84.5% after 800 cycles at 10.0 A g^−1^ (Figure 3f) [41]. Chen et al. synthesized a hollow-structured CoS_2_/nitrogen-doped carbon sphere composite derived from cobalt-based metal–organic frameworks. The void structure and carbon layer provide a buffer space for volume change and effectively inhibit the dissolution of polysulfides, ensuring the high electrical conductivity and structural stability of the composite. When used as an anode in SIBs, it delivered an impressive rate capability (637.4 mA h g^−1^ at 10 A g^−1^) [42].

##### Cobalt Selenide

The reaction mechanism of cobalt selenides (CoSe, CoSe_2_) is similar to iron selenides. There are similarities between different stoichiometric amounts of cobalt selenide, but the intermediate species will also be different during the chemical reaction. In the CoSe_2_ discharge process, Na_x_CoSe_2_ and CoSe are intermediate products, and in the fully discharged state, the products are Co and Na_2_Se, respectively. Similarly, CoSe forms the Na_x_CoSe mesophase. Zhang et al. prepared urchin-like single crystal CoSe_2_ assembled from many nanorods by a simple solvothermal method [43]. A systematic analysis of the sodium storage mechanism of CoSe_2_ was carried out by ex situ XRD. As shown in Figure 4a, firstly, the characteristic peak of CoSe appears when discharging from the open circuit voltage (1.3) to 0.9 V, and when located at 0.5 V, the physical phase of Na_2_Se is revealed, indicating the occurrence of the conversion reaction. After charging to 3.0 V, the Co K-edge position is the same as the CoSe_2_. Figure 4b further shows that Na^+^ ions are inserted into CoSe_2_ to form Na_x_CoSe_2_. Then, Na_x_CoSe_2_ reacts with Na^+^ ions to form CoSe and Na_2_Se. Finally, CoSe is converted to Co and Na_2_Se. The reaction mechanism of CoSe_2_ can be summarized as follows (Equations (13)–(17)):(13)$${\mathrm{CoSe}}_{2}{\;+\;\mathrm{xNa}}^{+}{\;+\;\mathrm{xe}}^{-}\to {\mathrm{Na}}_{x}{\mathrm{CoSe}}_{2}$$
(14)$${\mathrm{Na}}_{x}{\mathrm{CoSe}}_{2}\;+\;\left(2\;-\;x\right){\;\mathrm{Na}}^{+}{\;+\;(2\;-\;x)e}^{-}\to {\mathrm{CoSe}\;+\;\mathrm{Na}}_{2}\mathrm{Se}$$
(15)$${\mathrm{CoSe}}_{2}{\;+\;2\mathrm{Na}}^{+}{\;+\;2e}^{-}\to {\mathrm{Co}\;+\;\mathrm{Na}}_{2}\mathrm{Se}$$
(16)$$\mathrm{Co}\;+\;2\mathrm{NaSe}\to {\mathrm{Na}}_{x}{\mathrm{CoSe}}_{2}{\;+\;(4\;-\;x)\;\mathrm{Na}}^{+}{\;+\;(4\;-\;x)e}^{-}$$
(17)$${\mathrm{Na}}_{x}{\mathrm{CoSe}}_{2}\to {\mathrm{CoSe}}_{2}{\;+\;\mathrm{xNa}}^{+}{\;+\;\mathrm{xe}}^{-}$$

When CoSe_2_ is used as the anode material of sodium-ion battery, the discharge capacity still reaches 410 Ah g^−1^ after 1800 cycles, and it exhibits an excellent rate capability of 354 mA h g^−1^ at 10 A g^−1^. The introduction of carbon mechanism and the hybridization of atoms can further improve the electrochemical performance of CoSe_2_. For example, Zhang et al. successfully embedded CoSe_2_ nanoparticles into graphene nanovolumes (GNS) as anodes for sodium-ion batteries (Figure 4c,d). The CoSe_2_/C@GNS anodes exhibited a high reversible capacity of 545 mA h g^−1^ at 0.2 A g^−1^ and outstanding cycling stability of 455 mA h g^−1^ after 5000 cycles at 100 mA g^−1^. Moreover, CoSe_2_/C@GNS demonstrate ultra-high rate capabilities of 212.5 mA h g^−1^ at 50 A g^−1^ [44]. The uniform arrangement of nanoparticles and the unique network structure of CoSe_2_/C@GNS improve ion diffusion kinetics and cycle stability. Yang et al. successfully prepared necklace like-CNT/CoSe_2_@NC via the chemical method and selenization. MOFs are used as a source of doped carbon and are transformed into mesoporous CoSe_2_@NC nanoclusters of nanoparticles in the process of selenization, forming a CNT/CoSe_2_@NC composite with a necklace-like morphology [45]. The unique structure shortens the ion transmission path, and the close contact between the carbon nanotubes and CoSe_2_@NC effectively alleviates the dramatic change in volume during the recycling process. The CNT-CoSe_2_@NC exhibits a stable and reversible discharge capacity of 404 mA·h·g^−1^ after 120 cycles at 200 mA g^−1^. Huang et al. prepared a hierarchical porous Co_0.85_Se nanosheets as sodium-ion anodes by a simple hydrothermal method. Furthermore, they conducted a systematic study on the Na^+^ insertion/extraction mechanism, which can be described as follows [46] (Equation (18)):(18)$${\mathrm{Co}}_{0.5}{\mathrm{Se}+2\mathrm{Na}}^{+}{+\;2e}^{-}↔0.85{\mathrm{Co}\;+\;\mathrm{Na}}_{2}\mathrm{Se}$$

The hierarchical porous structure is conducive to fast electrolyte transport and the rapid adsorption of Na^+^ ions, while the synergistic effect of vacancy transfer and heterostructure between Co_0.85_Se and rGO improves conductivity and cycle stability. The Co_0.85_Se NSs@rGO electrode delivered specific rate capacities of 327 mA h g^−1^ at 4 A g^−1^ and a discharge capacity of 382 mA h g^−1^ after 800 cycles at 1 A g^−1^. Park and his colleagues used a two-step MOF-engaged approach to decompose and further selenize the Co-based zeolitic imidazolate framework precursor to prepare CoSe_2_@N-PGC/CNT [47].

Similarly, Tang et al. also used zeolite imidazole ester framework-67 (ZIF-67) particles as the carbon source to successfully synthesize hollow, carbon nanotube-connected carbon-coated CoSe_2_ nanospheres (CoSe_2_@C/CNTs) [48]. Jo et al. designed a unique nano-structured N-CNT/rGO/CoSe_2_ NF with a golden bristlegrass-like structure as a sodium-ion anode and showed good electrochemical performance. Reduced graphene oxide (rGO) matrix nanofibers are entangled with bamboo-like N-doped carbon nanotubes (CNT), and each node contains CoSe_2_ nanocrystals [49] Sun et al. prepared yolk–shell structured CoSe_2_/C nanospheres as multifunctional anode materials [50]. This structure has three-dimensional ion diffusion channels that can effectively hamper the aggregation of metal compound nanoparticles. Meanwhile, the CoSe_2_/C of the yolk–shell structure and a large number of pores help alleviate volume expansion and facilitate electrolyte wettability.

#### 2.1.3. Sb-Based Chalcogenides

##### Antimony Sulfides

Sb_2_S_3_ has a layered orthorhombic crystal structure and is an alloy-type anode material, and its sodium storage behavior involves sodium ion intercalation and conversion/alloying reactions. Each mole of Sb_2_S_3_ can accommodate 12 mol of sodium ions and can provide a high specific capacity (946 mA h g^−1^). Yao et al. synthesized carbon-coated Sb_2_S_3_ nanorods (SSNR/C) by the hydrothermal method as the anode material, and they studied the phase evolution behavior of SSNR/C by in situ transmission electron microscopy (Figure 5a,b). According to Figure 6b, Na_2_S, Na_3_Sb and Sb gradually disappeared with the continuous de-sodium treatment, which means that Na_2_S decomposition and Na_3_Sb dealloying occurred. When the sodium removal is complete, the transformation and alloying reactions between Sb_2_S_3_ and Na have excellent reversibility as analyzed by the corresponding SAED patterns and the corresponding radially integrated intensity distribution maps. The products corresponding to the intercalation reaction and conversion/alloying reaction of the SSNR/C electrode are amorphous Na_x_Sb_2_S_3_ and Na_3_Sb, Na_2_S, respectively during the reaction process. The reaction equation is as follows [51] (Equations (19) and (20)):

Na^+^ ion intercalation reaction:(19)$${\mathrm{Sb}}_{2}S_{3}{\;+\;\mathrm{xNa}}^{+}{\;+\;\mathrm{xe}}^{-}\to {\mathrm{Na}}_{x}{\mathrm{Sb}}_{2}S_{3}$$

Conversion/alloying reactions:(20)$${\mathrm{Na}}_{x}{\mathrm{Sb}}_{2}S_{3}\;+\;\left(12\;-\;x\right){\mathrm{Na}}^{+}{\;+\;(12\;-\;x)e}^{-}↔{2\mathrm{Na}}_{3}{\mathrm{Sb}\;+\;3\mathrm{Na}}_{2}S$$

The sodium storage behavior of Sb_2_S_3_ is dominated by conversion and alloying reactions, which leads to the problems of severe volume expansion and material crushing during the sodium intercalation process. In order to solve the above problems, Xiong et al. synthesized Sb_2_S_3_/SGS composites by rivets of nanostructured Sb_2_S_3_ on sulfur-doped graphene sheets (SGS) (Figure 5c).

When used as anode materials for sodium-ion batteries, they can maintain 83% stability after 900 cycles capacity retention. The porous structure of Sb_2_S_3_/SGS and the encapsulation of SGS in the outer layer can facilitate Na^+^ transport as well as adapt well to the volume changes during the sodiation/desodiation process to obtain high multiplicity and structural stability [52]. Yao et al. synthesized few-layered two-dimensional Sb_2_S_3_ (2D-SS) nanosheets by a scalable exfoliation method and applied them to anode materials for sodium-ion batteries.

The 2D-SS nanosheets with ultra-thin thickness (3.8 nm) and large surface area are rich in active centers and are able to shorten the ion diffusion path to ensure high-rate sodium ion transport. The 2D-SS nanosheets electrodes provide high rate capacities in SIBs (≈680 mA h g^−1^ at 0.05 A g^−1^) [53].

##### Antimony Selenides

Sb_2_Se_3_ is a semiconductor with a band gap of 1.17 eV. In the cyclic process, it follows a three-stage reaction: firstly, Na^+^ intercalates on Sb_2_Se_3_ to form the intermediate Na_x_Sb_2_Se_3_, after which Sb_2_Se_3_ is converted to Sb and Na_2_Se, and then, Sb_2_Se_3_ undergoes a metal alloying reaction, which can be expressed as Sb→NaSb→Na_x_Sb→Na_3_Sb. As shown in Figure 6a–c, Ou et al. reported that graphene-coated Sb_2_Se_3_ composites in the form of nanorods (Sb_2_Se_3_/rGO) were used as anode materials for SIBs, and the results of intercalation, conversion reactions and alloying reactions occurred during the cell operation were observed by in situ XRD. The reaction mechanism of Sb_2_Se_3_/rGO anode in different sodiation/desodiation states can be described as (Equations (21)–(26)):

Stage 1 (OCV–1.1 V, intercalation)
(21)$${\mathrm{Sb}}_{2}{\mathrm{Se}}_{3}{\;+\;\mathrm{xNa}}^{+}{\;+\;\mathrm{xe}}^{-}\to {\mathrm{Na}}_{x}{\mathrm{Sb}}_{2}{\mathrm{Se}}_{3}$$

Stage 2 (1.1–0.6 V, conversion)
(22)$${\mathrm{Na}}_{x}{\mathrm{Sb}}_{2}{\mathrm{Se}}_{3}\;+\;\left(6\;-\;x\right){\mathrm{Na}}^{+}{\;+\;(6\;-\;x)e}^{-}\to {2\mathrm{Sb}\;+\;3\mathrm{Na}}_{2}\mathrm{Se}$$

Stage 3 (0.6–0.01 V, alloying)
(23)$${\mathrm{Sb}\;+\;\mathrm{xNa}}^{+}{\;+\;\mathrm{xe}}^{-}\to {\mathrm{Na}}_{x}\mathrm{Sb}\;(x=1\sim 3)$$

Stage 4 (0.01–1.0 V, dealloying)
(24)$${\mathrm{Na}}_{x}\mathrm{Sb}\to {\mathrm{Sb}\;+\;\mathrm{xNa}}^{\;+\;}{\;+\;\mathrm{xe}}^{-}$$

Stage 5 (1.0–2.2 V, reversed conversion)
(25)$${2\mathrm{Sb}\;+\;3\mathrm{Na}}_{2}\mathrm{Se}\to {\mathrm{Na}}_{x}{\mathrm{Sb}}_{2}{\mathrm{Se}}_{3}\;+\;\left(6\;-\;x\right){\mathrm{Na}}^{\;+\;}{\;+\;(6\;-\;x)e}^{-}$$

Stage 6 (2.2–3.0 V, de-intercalation)
(26)$${\mathrm{Na}}_{x}{\mathrm{Sb}}_{2}{\mathrm{Se}}_{3}\to {\mathrm{Sb}}_{2}{\mathrm{Se}}_{3}{\;+\;\mathrm{xNa}}^{+}{\;+\;\mathrm{xe}}^{-}$$

The average mass specific energy capacity of Sb_2_Se_3_/rGO at a rate of 1.0 A g^−1^ after 500 cycles was 417 mA h g^−1^ with a capacity retention rate of 90.2% [53].

In order to solve the problem of poor structural stability of Sb_2_Se_3_, Xia et al. synthesized a dimensional carbon nanofiber Sb_2_Se_3_/CNF composite material by electrostatic spinning and selenization (Figure 6d,e). The unique nanostructure provides a large activity specific surface area and volume expansion space to ensure highly reversible and long period durability (532.6 mA h g^−1^ at 0.1 A g^−1^ after 120 cycles) [54]. Fang et al. synthesized PPy-coated Sb_2_Se_3_ (Sb_2_Se_3_@PPy) as an anode material for Na-ion batteries via the ion-exchange method. Due to the unique clip-like structure and PPy coating, Sb_2_Se_3_@PPy exhibits excellent rate performance (486 mA h g^−1^ at 2 A g^−1^), and no significant capacity degradation was observed even for Sb_2_Se_3_@PPy electrodes at a current density of 500 mA g^−1^ after 80 cycles [55].

#### 2.1.4. Zn-Based Chalcogenides

##### Zinc Sulfide

The sodium storage mechanism of ZnS has been studied extensively in recent years, and it was found that it also stores sodium ions through the combined effect of conversion and alloying reactions. Wei et al. synthesized necklace-like hollow ZnS-C nanofibers (ZnS@CNFs) by electrospinning combined with the solvothermal method (Figure 7a), and they investigated its sodium storage mechanism by ex situ XRD. As shown in Figure 7b, the corresponding ZnS characteristic peaks are essentially unchanged for discharge voltages from 2 to 0.7 V. From 0.7 to 0.25 V, the intensity of the ZnS characteristic peak decreases significantly, and the Na_2_S characteristic peak appears, indicating the start of the conversion reaction. When the discharge voltage was from 0.25 to 0.005 V, the characteristic peak of Na_2_S became the sharpest, which was accompanied by the appearance of the NaZn_13_ peak, indicating the conversion reaction is completed at this moment and the alloying reaction between Zn and Na^+^ appears. During the subsequent charging process, Zn undergoes a conversion reaction to ZnS at 0.005–1.3 V. When charged to 3.0 V, only the diffraction peaks of ZnS appear, indicating that ZnS@CNFs have good reversibility during the charge/discharge process. The reaction equation is as follows [56] (Equations (27) and (28)):(27)$${13\mathrm{Zn}\;+\;3\mathrm{Na}}^{+}{\;+\;3e}^{-}↔{3\mathrm{NaZn}}_{13}$$
(28)$${\mathrm{ZnS}+2\mathrm{Na}}^{+}{\;+\;2e}^{-}↔{\mathrm{Zn}+\mathrm{Na}}_{2}S$$

Li et al. synthesized ZnS/NCF composites with a porous fibrous shape in order to solve the problem of huge volume expansion caused by the alloying reaction on pure ZnS (Figure 7c). The ZnS/NCF composite as an anode material for sodium-ion batteries provides a specific capacity of 455 mA h g^−1^ at 0.1 A g after 50 cycles. The problem of volume expansion is well solved by the porous one-dimensional structure [57]. Jing et al. prepared N and S co-doped ZnS/NSC composites by the direct calcination of zinc sulfide. As an anode for SIBs, the ZnS/NSC exhibits excellent stability (500.7 mA h g^−1^ at 500 mA g^−1^, ≈96.9% capacity retention after 100 cycles) and rate performance (364.9 mA h g^−1^ at 800 mA g^−1^) (Figure 7d) [58]. As shown in Figure 7e,f, Li et al. successfully synthesized ZnS/NPC with a nitrogen-doped porous structure by a co-precipitation method combined with a simple carbonization and sulfidation process for sodium ion battery anode materials, and the resulting ZnS/NPC composites exhibited a capacity of 289.2 mA h g^−1^ after 1000 cycles at 1 A g^−1^. The nitrogen-doped porous carbon polyhedron can alleviate the severe volume expansion of the material during the alloying process and improve the electrical conductivity of the material [59].

##### Zinc Selenides

ZnSe has a diamond-like structure and exhibits better electrical conductivity due to its smaller band gap (2.7 eV). ZnSe and ZnS have similar storage mechanisms (initial conversion reactions and alloying reactions). Liu et al. successfully synthesized layered ZnSe@HCNs composites by attaching ultra-fine ZnSe nanoparticles to the inner and outer surfaces of hollow carbon nanospheres through a solvothermal method and investigated the phase transition of ZnSe during charging/discharging by ex situ XRD (Figure 8a,b). The phase change involved in the reaction process can be expressed by the following equation (Equations (29) and (30)):(29)$${\mathrm{ZnSe}+2\mathrm{Na}}^{+}{+\;2e}^{-}↔{\mathrm{Na}}_{2}\mathrm{Se}+\mathrm{Zn}$$
(30)$${13\mathrm{Zn}+\mathrm{Na}}^{+}{+\;e}^{-}↔{\mathrm{NaZn}}_{13}$$

The unique hollow layered hybrid nanostructure of ZnSe@HCN maintains a capacity of 361.9 mA h g^−1^ after 1000 cycles at 1 A g^−1^ and provides a high reversible capacity of 266.5 mA h g^−1^ even after 1000 cycles at high currents of 20 A g^−1^ [60].

He et al. used ZIF-8 as the carbon source and decorated ZnSe nanoparticles on a carbon framework of nitrogen-doped hollow polyhedra to prepare ZnSe NP@NHC composites (Figure 8c). The uniformly distributed ZnSe provided abundant active sites, which effectively enhanced the electrical conductivity and improved the charge transfer kinetics of the material. It has a reversible capacity of 251 mAh g^−1^ at 0.2 A g^−1^ after 500 cycles and can provide a multiplicative capacity of 225 mA h g^−1^ at 3 A g^−1^ (Figure 8d) [61]. Dong et al. designed a willow-leaf-like nitrogen-doped carbon-coated ZnSe (ZnSe@NC) composite, as shown in Figure 8e. The unique structure allows ZnSe@NC to exhibit excellent cycling stability performance as a sodium ionization anode material (242.2 mA h g^−1^ at 8.0 A g^−1^ even after 3200 cycles) (Figure 9f) [62].

#### 2.1.5. Cu-Based Chalcogenides

##### Copper Sulfide

CuS, Cu_2_S and Cu_9_S_5_ have been used as potential anode materials for sodium ion batteries. CuS has attracted much attention due to its high copper mobility, and its sodium storage mechanism has been clearly elucidated. Yang et al. prepared hydrangea-shaped CuS microspheres and verified by SRD and in situ high-resolution synchrotron diffraction analysis that CuS undergoes intercalation and conversion reactions during the charge storage process (Figure 9a,b). During the reaction, CuS rapidly undergoes disproportionation to form Cu_2_S intermediates, which was finally converted to Cu and Na_2_S [63]. The reaction equation is as follows (Equations (31)–(33)):(31)$${\mathrm{CuS}+\mathrm{xNa}}^{+}{\;+\;\mathrm{xe}}^{-}\to {\mathrm{Na}}_{x}\mathrm{CuS}\;(x\;<\;2)$$
(32)$${\mathrm{Na}}_{x}\mathrm{CuS}\to (x/{2)\mathrm{Cu}}_{2}S+(x/{2)\mathrm{Na}}_{2}S\;+\;\left(1-x\right)\mathrm{CuS}$$
(33)$${\mathrm{Cu}}_{2}{S+2\mathrm{Na}}^{+}{+\;2e}^{-}↔{\mathrm{Na}}_{2}S+2\mathrm{Cu}$$

It exhibits good cycling stability and excellent rate performance (350 mA h g^−1^ at 0.1 A g^−1^) when used as an anode material for Na-ion batteries. Sun synthesized a flexible free-standing anode composed of flower-like copper sulfide (CuS), metallic copper (Cu) nanoparticles, and multi-walled carbon nanotubes (CNTs) using a simple electro-sulfurization method (Figure 9c). Carbon nanotubes and those copper particles trapped in the carbon nanotube network provide a highly conductive matrix for electrons. The CuS-Cu@CNTs electrode exhibits a reversible capacity of 512.5 mA h g^−1^ after 1100 cycles at 2400 mA g^−1^, corresponding to a high capacity retention rate of 87.5% [64]. Wu et al. used an ion exchange process to obtain hollow CuS as the anode material for Na-ion batteries (Figure 9d,e). The special hollow structure solves the problem of large volume expansion during the intercalation of sodium ions, and it exhibits excellent cycle performance (the charge capacity of copper sulfide remains at 361 mA h g^−1^ at 100 mA g^−1^ after 100 cycles) [65].

### 2.2. Layer-Structured MCs

#### 2.2.1. Mo-Based Chalcogenides

##### Molybdenum Sulfide

Molybdenum disulfide is one of the most important members of MCs with a typical graphite-like layered structure. The unique interlayer is connected by van der Waals forces, the interlayer is dominated by covalent bonds, and the interlayer distance is 0.62 nm. Compared with graphite (0.335 nm), molybdenum disulfide has a larger interlayer spacing, so the reaction kinetics has a better performance. MoS_2_ has received extensive attention due to its satisfactory sodium storage performance in the field of sodium-ion battery anodes. Wang et al. synthesized a layered hollow tube assembled from molybdenum disulfide nanosheets by a simple hydrothermal method (Figure 10a). As shown in Figure 10b, its structure and phase variations during cycling were investigated by in situ XRD [66]. During the first discharge, when the discharge potential changed from 0.9 to 0.4 V, the (002) peak of MoS_2_ shifted to 12.1°, which was attributed to the insertion of Na^+^ into MoS_2_ to form Na_x_MoS_2_. When the discharge potential changed from 0.4 to 0.05 V, the presence of the diffraction peak of Na_2_S could be observed. During charging, the diffraction peak corresponding to MoS_2_ reappears, indicating the presence of a good reversible reaction. The initial discharge mechanism equation of MoS_2_ in the use of sodium ion batteries is as follows (Equations (34) and (35)):(34)$${\mathrm{MoS}}_{2}\;{+\;\mathrm{xNa}}^{+}\;{+\;\mathrm{xe}}^{-}↔{\mathrm{Na}}_{x}{\mathrm{MoS}}_{2}\;(x\;<\;1)$$
(35)$${\mathrm{Na}}_{x}{\mathrm{MoS}}_{2}+\left(4-x\right){\mathrm{Na}}^{+}\;+\;(4-{x)e}^{-}↔{\mathrm{Mo}+2\mathrm{Na}}_{2}S$$

The resulting tubular MoS_2_ has a high specific capacity of 652.5 mA h g^−1^ after 50 cycles at a current density of 100 mA g^−1^. Molybdenum sulfide as a sodium ion anode still has serious circulation problems and large voltage polarization. Many methods to solve the inherent problems of MoS_2_ have been reported. For example, Zhao et al. used a simple vacuum infiltration and heating process to fabricate a super-elastic three-dimensional multi-layer MoS_2_/carbon frame heterogeneous electrode (MoS_2_@CF) (Figure 10c,d). MoS_2_@CF with a three-dimensional heterogeneous network of layered pores has excellent flexibility, and the electrode shows excellent cycle stability with high reversible capacities reaching up 240 mA h g^−1^ after 500 cycles at 1 A g^−1^ and high rate capacities (171 mA h g^−1^ at 5 A g^−1^) [67]. Hou et al. successfully synthesized hollow layered MoS_2_@NHCS composites using N-doped hollow carbon nanospheres (NHCS) as a template and then modified them with MoS_2_ nanosheets (Figure 10e,f). The NHCS effectively slowed down the structural collapse of the composite during cycling and promoted the ion diffusion and charge transfer rate. When used as SIBs anode, the capacity of MoS_2_@NHCS was 371 mA h g^−1^ at 1 A g^−1^, and the capacity retention rate was 94.9% after 100 cycles [68]. Luo et al. obtained SPAN-Mo-475 composites by embedding MoS_2_ nanocrystals and sulfur nanodots simultaneously in sulfated polyacrylonitrile (SPAN) fibers by the electrospinning technique. The MoS_2_ nanoparticles and sulfur nanodots uniformly anchored within the SPAN nanofibers provided abundant active sites and shortened ion transport channels, which contributed significantly to the fast kinetics. In addition, the sulfur defects in MoS_2_ are well resolved by the sulfur dissolution shuttle effect. Therefore, the electrode exhibits an excellent high reversible capacity (326 mA h^−1^ at 5 A after 600 cycles, even at 10 A can exhibit 214 mA h g^−1^ after 600 cycles) [69]. Zhang et al. designed a composite material with a special structure, which is composed of molybdenum disulfide (MoS_2_) nanosheets arranged vertically on a carbon paper derived from paper towels. At a current density of 20 mA g^−1^, a high reversible capacity of 446 mA h g^−1^ can be achieved [70]. Wang et al. successfully prepared MoS_2_ hollow spheres composed of many curved nanosheets via a one-pot hydrothermal method. The MoS_2_ with the advantages of hollow structure shows superior rate performance (347.3 mA h g^−1^ at 0.5 A g^−1^) and long cycle performance (334.6 mA h g^−1^ at 2 A g^−1^ after 1200 cycles) [71]. Yuan and his colleagues developed Nb_2_CT_x_ MXene-framework MoS_2_ nanosheets coated with carbon (Nb_2_CT_x_@MoS_2_@C) and constructed a robust three-dimensional cross-linked structure. In the design of Nb_2_CTx@MoS_2_@C, the layered carbon coating and highly conductive Nb_2_CT_x_ MXene nanosheets effectively solve the problem of volume expansion and improve the diffusion efficiency and promote fast dynamics. The Nb_2_CT_x_@MoS_2_@C negative electrode provides an ultra-high reversible capacity of 530 mA h g^−1^ after 200 cycles at 0.1 A g^−1^ [72]. Li et al. applied the interface engineering strategy to construct a MoS_2_/C composite material (MoS_2_-C@C) with an overlapping hierarchical structure through a bottom–up synthesis method [73]. The alternative stacking of molybdenum disulfide and carbon layers constitutes a reasonable structure, and the heterogeneous interface in MoS_2_-C@C provides a rich electron transfer path, so MoS_2_-C@C exhibits the excellent rate capability of 164 mA h g^−1^ at 20 A g^−1^. Wang et al. rationally designed and synthesized a hierarchical core/shell nanoarchitecture composed of 1D NTO nanowires core and MoS_2_ nanosheets coupled with carbon nanosheets. The NTO/MoS_2_-C exhibits ultra-durable cycle capability. During ultra-fast charge and discharge within 80 s, the capacity can also be maintained at 201 mA h g^−1^ after 16,000 cycles [74].

##### Molybdenum Selenide

The MoSe_2_ has a layered structure similar to MoS_2_. However, the MoSe_2_ has a larger interlayer spacing (0.646 nm) and better electrical conductivity. Zheng et al. used a hydrothermal method to develop three-dimensional (3D) flower-like MoSe_2_/CN composites for sodium-ion battery anode materials, and they investigated the sodium storage mechanism of MoSe_2_ by in situ XRD. As shown in Figure 11a,b, when the discharge voltage is from the open circuit voltage (OCV) to 0.8 V, the intensity of the diffraction peak corresponding to MoSe_2_ gradually decreases, indicating that the intercalation reaction occurs, and Na^+^ ions are intercalated into MoSe_2_ to form the intermediate Na_x_MoSe_2_. When fully discharged to 0.01 V, the characteristic peak of MoSe_2_ disappears and Na_2_Se appears, indicating that Na_x_MoSe_2_ and Na^+^ have undergone a conversion reaction, and the products are Na_2_Se and metal Mo. The peak of Na_x_MoSe_2_ was observed when the charging voltage was from 0.01 to 1.5 V. The diffraction peak corresponding to MoSe_2_ was re-detected after full charging to 3.0 V, indicating a good reversible response of MoSe_2_ in the cycle. The reaction mechanism of MoSe_2_ is as follows [75] (Equations (36)–(39)):

At the discharge process:(36)$${\mathrm{MoSe}}_{2}{\;+\;\mathrm{xNa}}^{+}{\;+\;e}^{-}\to {\mathrm{Na}}_{x}{\mathrm{MoSe}}_{2}\;(\mathrm{OCV}–0.08\;V)$$
(37)$${\mathrm{Na}}_{x}{\mathrm{MoSe}}_{2}{\;+\;\mathrm{Na}}^{+}{\;+\;e}^{-}\to {\mathrm{Na}}_{2}\mathrm{Se}+\mathrm{Mo}\;(0.8–0.01\;V)$$

At the charge process:(38)$${\mathrm{Na}}_{2}\mathrm{Se}+\mathrm{Mo}\to {\mathrm{Na}}_{x}{\mathrm{MoSe}}_{2}{\;+\;\mathrm{Na}}^{+}{\;+\;e}^{-}\;\left(0.01–1.5\;V\right)$$
(39)$${\mathrm{Na}}_{x}{\mathrm{MoSe}}_{2}\to {\mathrm{MoSe}}_{2}{\;+\;\mathrm{Na}}^{+}{\;+\;e}^{-}\;\left(1.5–3.0\;V\right)$$

When tested as anode for SIBs, the obtained MoSe_2_/CN exhibited a reversible capacity of 328.7 mA h g^−1^ after 500 cycles at 1.0 A g^−1^ (Figure 11c,d). MoSe_2_ as an electrode is still limited by problems such as low conductivity and poor stability. Through the modification of bare MoSe_2_, such as carbon coating, porous structure design and nanometerization, its excellent electrochemical performance can be achieved. As shown in Figure 11e, Muhammad et al. proposed a 3D three-layer design in which MoSe_2_ is sandwiched between the inner carbon nanotube (CNT) core and the outer carbon layer by solvothermal carbonization to form a CNT/MoSe_2_/C framework.

The 3D porosity of the heterostructure remains intact after an intense densification process to produce a high areal capacity of 4.0 mA h cm^−2^ and a high mass loading of 13.9 mg cm^−2^ with a gravimetric capacity of 347 mA h g^−1^ at 500 mA g^−1^ after 500 cycles. CNT/MoSe_2_/C exhibits excellent rate capability (307 mA h g^−1^ at 1600 mA g^−1^) (Figure 11f) [76]. Liu et al. successfully synthesized MoSe_2_@hollow carbon nanosphere (HCNS) material with an encapsulated structure. Hollow carbon nanospheres effectively promote the formation of several layers of MoSe_2_ nanosheets with a uniformly restricted interlayer spacing structure. MoSe_2_@HCNS as the anode of sodium ion battery shows good electrochemical performance with a discharge capacity of 254.4 mA h g^−1^ at 5 A g^−1^ after 1800 cycles and the rate capabilities of 382 mA h g^−1^ at 10 A g^−1^ [77]. Ge et al. successfully prepared nanospheres composed of PVP-coated MoSe_2_ nanosheets. The MoSe_2_/N-C showed a reversible capacity of 552.1 mA h g^−1^ at 0.1 A g^−1^ over 120 cycles [78]. Zhang et al. synthesized porous MoSe_2_/C nanofibers with a one-dimensional carbon composite nanostructure with uniformly distributed nanoparticles by electrospinning. Porous MoSe_2_/C composite nanofibers show excellent electrochemical performance. The discharge capacity is 454 mA h g^−1^ at 0.2 A g^−1^ after 200 cycles [79]. Zhang et al. controlled the growth of oriented, interlayer-expanded MoSe_2_ nanosheets (MoSe_2_/G) on graphene with Mo-C bonds through a hydrothermal reaction. The graphene effectively controls the rapid growth and clustering of MoSe_2_, and it promotes the transfer of electrons and sodium ions on the interface and the reversible insertion/extraction of sodium ions, thus exhibiting good cycle performance [80]. Zhang et al. designed a 3D spatial structure-like MoSe_2_@MPCS by using Ni-MOF as a self-sacrificing template precursor. The three-dimensional structure not only provides an excellent conductivity environment but also effectively avoids the rapid re-stacking of MoSe_2_ and ensures the structural integrity of MoSe_2_@MPCS during the reaction. The MoSe_2_@MPCS electrode exhibits excellent electrochemical performance with a discharge capacity of 254.4 mA h g^−1^ at 5 A g^−1^ after 1800 cycles [81].

#### 2.2.2. V-Based Chalcogenides

##### Vanadium Sulfide

Vanadium has abundant valence states, and its sulfides include VS_2_, VS_4_, and V_5_S_8_. The most common type is VS_2_, which exhibits a layered structure with a spacing of 5.76 nm. The layered structure and metallic properties of VS_2_ are one of the important factors to provide high rate performance. Yu successfully fabricated hierarchical flower-like VS_2_ nanosheet assemblies by the solvothermal method and investigated their sodium storage mechanism (Figure 12a–c). Flower-like VS_2_ provides a high reversible capacity of about 600 mA h g^−1^ after 50 cycles at 0.1 A g^−1^ as anode for Na-ion batteries. They then studied the Na^+^ storage and release mechanism of the material by ex situ Raman (Figure 12a). The research and analysis confirmed that the material retained a 2D layered structure at a low Na^+^ insert into the VS_2_ layers from Na_x_VS_2_ and then converted to a NaVS_2_ nanocomposite during deeper sodiation. The reaction equation is as follows: [82] (Equation (40)).
(40)$${\mathrm{VS}}_{2}{\;+\;2\mathrm{Na}}^{+}{\;+\;2e}^{-}↔{\mathrm{Na}}_{2}{\mathrm{VS}}_{2}$$

Sun et al. synthesized a multilayered nanosheet-stacked VS_2_ (VS_2_-SNS) via a facile one-step polyvinylpyrrolidone (PVP)-assisted assembly method (Figure 12d,e). The highly stable layered structure provides excellent electrochemical performance of VS_2_ when used as an anode material for sodium ion batteries (204 mA h g^−1^ at 5 Ag^−1^ after 600 cycles). They analyzed the electrochemical kinetics of VS_2_ and showed that the sodium ion charge storage depends on the intercalation pseudocapacitance behavior with a capacitive contribution of up to 69% of the total capacity at 1 mV s^−1^ [83]. To further improve the sodium storage properties of VS_2_, Liu et al. proposed the solution of constructing a three-dimensional assembled structure. They have controllably synthesized three-dimensional layered VS_2_ microrods assembled from nanosheets composed of VS_2_ small particles by the in situ chemical etching method (Figure 12f,g). When applied as an anode for sodium ion batteries, this material exhibited excellent multiplicative performance (rate capacity of 323 mA h g^−1^ at 0.2 mA g^−1^) (Figure 12h) [84]. VS_4_ and V_5_S_8_ are non-layered structures that are different from the layered structure of VS_2_, but it was found that VS_4_ and V_5_S_8_ also exhibit excellent electrochemical performance in sodium-ion storage. For example, Sun synthesized VS_4_/rGO composites and verified the presence of intercalation and conversion reactions during sodium storage by in situ XRD. The VS_4_/rGO composite was converted from VS_4_ to metal V and Na_2_S during the sodiation process, which further verified the existence of the conversion reaction. VS_4_/rGO provides a rate capacity of 362 mA h g^−1^ at 100 mA g^−1^ when used as an anode material. The reaction mechanism of V_5_S_8_ is similar to VS_4_, and it exhibits impressive sodium storage properties. Liu et al. synthesized V_5_S_8_@CNF by anchoring V_5_S_8_ nanoparticles on carbon nanofibers. It exhibits excellent performance after 400 cycles at 0.2 A g^−1^, with a specific capacity of 351 mA h g^−1^ when used as an anode material. Such excellent performance is attributed to the abundant active sites provided by V_5_S_8_ uniformly distributed on the CNF [85].

##### Vanadium Selenide

Compared with vanadium sulfide, vanadium selenide has a similar crystal structure and metallic features, and it has higher electrical conductivity (1 × 10^3^ S m^−1^). Wu synthesized VSe_2_/NCNFs composites by encapsulating ultra-fine VSe_2_ particles in N-doped carbon nanofibers and investigated its sodium storage mechanism. Nanoparticle-like VSe_2_ and highly graphitized carbon fibers provide fast transport channels for ions and electrons and effectively alleviate the volume expansion of VSe_2_ during cycling. VSe_2_/NCNFs composite as a sodium-ion anode material provides a high reversible capacity of 420.8 mA h g^−1^ at 0.05 A g^−1^ after 10,000 cycles [86].

#### 2.2.3. W-Based Chalcogenides

##### Tungsten Sulfide

Tungsten and molybdenum are in the same main group, and they exhibit similar physicochemical and structural properties (hexagonal phase). When WS_2_ is used as the anode material for Na-ion batteries, the reaction mechanism is divided into the intercalation process and the conversion reaction, the latter being the main contributor to the sodium storage behavior. Hu et al. fabricated a 3D layered hollow microflower bud-like hybrids composite (H-WS_2_@NC) by a solution thermal method. As shown in Figure 13a,b, the reaction mechanism of the H-WS_2_@NC electrode was further investigated by in situ XRD. When the electrode was discharged to 0.6 V, two diffraction peaks with broadening can be clearly observed, which was an indication of the formation of Na_x_WS_2_ by Na^+^ embedded in the WS_2_ layer. The reaction equation was as follows in Equation (41). When the electrode is fully discharged to 0.01 V, there is only one new diffraction peak at 39°, which proves that Na_x_WS_2_ was completely converted into W and Na_2_S, implying the end of the conversion reaction. The corresponding equation was represented by Equation (42). During the charging process, the peak of Na_2_S still exists, but it weakens at 1.8 V, indicating the Na_2_S was partially converted Equation (43). When charged to 3.0 V, the peak of Na_2_S disappears and is replaced by two peaks of WS_2_, indicating that W and Na_2_S undergo a reverse conversion reaction to form WS_2_ (Equations (41)–(44)) [87].
(41)$${\mathrm{WS}}_{2}{\;+\;\mathrm{xNa}}^{+}{\;+\;\mathrm{xe}}^{-}\to {\mathrm{Na}}_{x}{\mathrm{WS}}_{2}$$
(42)$${\mathrm{Na}}_{x}{\mathrm{WS}}_{2}\;+\;\left(4\;-\;x\right){\mathrm{Na}}^{+}{\;+\;(4\;-\;x)e}^{-}\to {2\mathrm{Na}}_{2}S\;+\;W$$
(43)$${\mathrm{Na}}_{2}S\to {S\;+\;2\mathrm{Na}}^{+}{\;+\;2e}^{-}$$
(44)$${2\mathrm{Na}}_{2}S+W\to {\mathrm{WS}}_{2}{\;+\;4\mathrm{Na}}^{+}{\;+\;4e}^{-}$$

The 3D H-WS_2_@NC exhibited high and stable reversible capacity (375 mA h g^−1^ at 1.0 A g^−1^ after 1000 cycles) (Figure 13c). Song et al. prepared honeycomb WS2/rGO Nano-HC composites by self-assembly of WS_2_ nanoparticles with graphene using a hydrothermal method (Figure 13d,f). The graphene-supported nano honeycomb planar structure has a larger specific surface area and better electrical conductivity. Therefore, WS_2_/rGO Nano-HC exhibited good sodium ion storage properties (the charge capacity remained at 630.5 mAhg^−1^ at 100 mAg^−1^ after 100 cycles) (Figure 13e) [88]. Li et al. formed WS_2_@S/N-C composites by embedding WS_2_ nanosheets into lotus rhizome-like carbon nanofibers (Figure 13g). As shown in Figure 13h,i, the WS_2_@S/N–C nanofibers exhibit an excellent rate capacity of 108 mA h g^−1^ at 30 A g^−1,^ Moreover, the discharge capacity of WS_2_@S/N–C reached 319 mA h g^−1^ at 0.1 A g^−1^ after 100 cycles (Figure 13i) [89]. The synergistic effect between WS_2_ nanosheets and carbon layers expands the interlayer spacing and provides a large channel for sodium ion insertion. In addition, the lotus root-like nanofiber framework not only provides good electrical conductivity but also effectively reduces the volume expansion during the reaction.

##### Tungsten Selenide

Tungsten selenide has received extensive attention from researchers due to its good electrical conductivity and large lattice parameter. Kim et al. successfully fabricated C@WSe_2_@PCC composites by uniformly assembling well-crystallized WSe_2_ nanosheets on porous carbon cloth. The C@WSe_2_@PCC electrode exhibited outstanding electrochemical performances (1.05 mA h cm^−2^ at 0.75 mA cm^−1^ after 150th cycles) [90]. Cho et al. prepared WSe_2_–rGO composites by spray pyrolysis combined with the selenization process. The discharge capacity of WSe_2_-rGO for sodium ion storage is 238 mA h g^−1^ at 0.5 A g^−1^ after the 100th cycle [91].

#### 2.2.4. Ni-Based Chalcogenides

##### Nickel Sulfide

The presence of multivalent states of nickel can form NiS with hexagonal, NiS_2_ with cubic shape and Ni_3_S_2_ compounds with rhomboidal structure with sulfur. The charge storage of NiS and NiS_2_ is accomplished by intercalation and conversion reactions. Fan et al. successfully prepared Ni@NCNTs composites with a core–shell structure by in situ chemical transformation and then used in situ XRD to reveal the chemical reactions of NaS in the process of sodium storage. Ni@NCNTs retained a stable cycling capacity of 89.3% at 1 A g^−1^ after 500 cycles when used as an anode electrode material. As shown in Figure 14a,b, the main NiS peaks located at 30.2, 34.8 and 45.9 are clearly shifted to the right during the process from the open circuit voltage to 1.2 V. This phenomenon is the result of the insertion of Na ions and the formation of Na_x_NiS. The voltage from 1.2 to 0.95 V is accompanied by the disappearance of the Na_x_NiS peak and the gradual formation of the peaks of Ni_3_S_2_ and Na_2_S. When discharged to 0.01 V, only diffraction peaks corresponding to N and Na_2_S appear in the XRD pattern, indicating the completion of the corresponding conversion reaction during the sodiation process. The gradual decrease in the peak intensities corresponding to Na_2_S and Ni during the desodiation process (0.01–1.7 V) can be attributed to the occurrence of the conversion reaction that converts Ni to amorphous Ni_3_S_2_. After further charging to 3.0 V, the peaks of Ni and Na_2_S disappeared completely. The chemical equation for the whole process is as follows [92] (Equations (45)–(48)):(45)$${\mathrm{NiS}\;+\;\mathrm{xNa}}^{+}\;{+\;\mathrm{xe}}^{-}\to {\mathrm{Na}}_{x}{\mathrm{NiS}}_{2}$$
(46)$${3\mathrm{Na}}_{x}{\mathrm{NiS}}_{2}+\left(2\;-\;3x\right){\mathrm{Na}}^{+}+\left(2\;-\;3x\right)e^{-}\to {\mathrm{Na}}_{2}{S+\mathrm{Ni}}_{3}S_{2}$$
(47)$${\mathrm{Ni}}_{3}S_{2}{\;+\;4\mathrm{Na}}^{+}{\;+\;4e}^{-}\to {2\mathrm{Na}}_{2}S\;+\;3\mathrm{Ni}$$
(48)$${3\mathrm{Ni}\;+\;2\mathrm{Na}}_{2}S\to {\mathrm{Ni}}_{3}S_{2}{\;+\;4\mathrm{Na}}^{\;+\;}{+\;4e}^{-}$$

NiS_2_ has a similar reaction mechanism to NiS. Zhao et al. prepared uniformly distributed NiS_2_ nanosheets on porous carbon microtubes (NiS_2_/pCMT) as an anode material. The high chemical affinity of the NiS_2_ nanosheet surface for sodium ions was confirmed by first-principles density flooding theory (DFT) calculations (Figure 14c). Moreover, the energy barrier reveals that the NiS_2_ (100) surface shows a lower energy barrier than that of the NiS_2_ (111) surface. The NiS_2_/pCMT provided a high energy density of 136 Wh kg^−1^ when used as an anode material for sodium-ion batteries [93].

To verify that Ni_3_S_2_ has excellent sodium storage properties, Zhou et al. prepared Ni_3_S_2_/CNTs composites by the hydrothermal method and explored the reaction mechanism by ex situ XRD (Figure 14d,e). When at 0.59 V in the discharged state, the characteristic peaks of Ni_3_S_2_ disappeared, while those of Na_2_S and Ni appeared. When in the discharged state (0.01 V), Ni_3_S_2_ has been completely converted to Na_2_S and Ni. During charging (0.01–3.00 V), the intensity of Na_2_S and Ni peaks weakened, while the peaks located in Ni_3_S_2_ enhanced, indicating that the Ni_3_S_2_/CNTs composites have good reversibility. The charge capacity retention of the Ni_3_S_2_/CNTs electrode was up to 82.2% at 0.5 A g^−1^ after 200 cycles [94].

##### Nickel Selenide

NiSe_2_ has a very similar structure and reaction mechanism to NiS_2_ and has a theoretical capacity of 495 mA h g^−1^. The intermediate products of the discharge process are Na_x_NiSe_2_ and NiSe, respectively, and the final products are Ni and Na_2_Se. During the charging process, Na_2_Se and Ni are gradually lost and then completely converted to NiSe_2_. Ou et al. prepared NiSe_2_/rGO composites by the hydrothermal method and investigated the chemical reactions occurring when NiSe_2_/rGO was used as an anode material for sodium-ion batteries by in situ XRD. The reaction equations are as follows [95] (Equations (49)–(52)):(49)$$\mathrm{OCV}–1.0\;V:\;\;\;\;{\mathrm{NiSe}}_{2}{\;+\;\mathrm{xNa}}^{+}{\;+\;e}^{-}\to {\mathrm{Na}}_{x}{\mathrm{NiSe}}_{2}$$
(50)$$1.1–0.4\;V:\;\;\;\;{\mathrm{Na}}_{x}{\mathrm{NiSe}}_{2}{\;+\;\mathrm{Na}}^{+}{\;+\;e}^{-}\to {\mathrm{Na}}_{2}\mathrm{Se}+\mathrm{Ni}$$

In the charge process:(51)$$0.4–1.9\;V:\;\;\;\;{\mathrm{Na}}_{2}\mathrm{Se}\;+\;\mathrm{Ni}\to {\mathrm{Na}}_{x}{\mathrm{NiSe}}_{2}{\;+\;\mathrm{Na}}^{+}{\;+\;e}^{-}$$
(52)$$1.9–3.0\;V:\;\;\;\;{\mathrm{Na}}_{x}{\mathrm{NiSe}}_{2}\to {\mathrm{NiSe}}_{2}{\;+\;\mathrm{Na}}^{+}{\;+\;e}^{-}$$

Ge et al. successfully prepared microspheres of carbon-constrained NiSe_2_ by selenization of the precursor (Ni-precursor/PPy). The layered hollow structure and the double N-doped carbon layer allow the material to exhibit excellent cycling performance (374 mA h g^−1^ at 10.0 A g^−1^ after 3000 cycles) [96].

#### 2.2.5. Sn-Based Chalcogenides

##### Tin Sulfide

Tin sulfides are generally present in the form of SnS_2_ and SnS. They exhibit square (SnS) and hexagonal (SnS_2_) structures, and their interlayer spacings are 0.433 and 0.590 nm, respectively. Both SnS and SnS_2_ undergo similar conversion and alloying reactions during sodiation/desodiation. As shown in Figure 15a, Zhou et al. fabricated SnS_2_ (SnS_2_ NWAs) with a unique nanowall arrays structure by vapor deposition, and they achieved a high reversible capacity of 576 mA h g^−1^ at 500 mA g^−1^ when used as an anode material. Thus, it was verified that SnS_2_ has a good energy storage value [97]. Cui synthesized SnS_2_/graphene-carbon nanotube aerogel (SnS_2_/GCA) composites and investigated the reaction mechanism of SnS_2_ by in situ TEM, and the results are shown in Figure 15b. The reaction equation was as follows [98] (Equations (53)–(55)):(53)$${\mathrm{SnS}}_{2}{\;+\;\mathrm{Na}}^{\;+\;}{\;+\;e}^{-}↔{\mathrm{NaSnS}}_{2}$$
(54)$${\mathrm{NaSnS}}_{2}{\;+\;3\mathrm{Na}}^{+}{\;+\;3e}^{-}↔{\mathrm{Sn}\;+\;2\mathrm{Na}}_{2}S$$
(55)$${4\mathrm{Sn}\;+\;15\mathrm{Na}}^{+}{\;+\;15e}^{-}↔{\mathrm{Na}}_{15}{\mathrm{Sn}}_{4}$$

However, Na^+^ is more easily alloyed with SnS to form Na_x_Sn, such that SnS has a higher intercalation capacity and structural stability than SnS_2_. The sodium storage mechanism of SnS still suffers from problems such as material crushing caused by Na^+^ insertion. To mitigate these problems, Xue et al. synthesized three-dimensional porous SnS/C composites using silica opal as a template (Figure 15c). The homogeneous distribution of SnS nanoparticles on 3D porous carbon interconnect structures provides an important guarantee for the high electrical conductivity and structural integrity of the composites. As the anode of SIBs, SnS/C exhibits a specific capacity of 400 mA h g^−1^ at 100 mA g^−1^ after 100 cycles [99].

##### Tin Selenide

Compared with SnS and SnS_2_, SnSe and SnSe_2_ have higher electrical conductivity and mass specific capacity, showing great prospects in the application of sodium-ion batteries. The charge storage mechanism of SnSe and SnSe_2_ is similar to that of SnS (initial intercalation reaction, sequential transformation and alloying reaction). Wang et al. synthesized a sandwich-like SnSe_2_/reduced graphene oxide (rGO) composite by a simple one-pot solvothermal technique, and they investigated its sodium storage mechanism by ex situ XRD. There was initial sodiation to form intermediate Na_x_SnSe_2_, which was followed by transformation and alloying reaction to form Na_15_Sn_4_, Na_29.58_Sn_8_, Sn, and Na_2_Se, and finally complete discharge to form Na_15_Sn_4_ [100]. Liu et al. loaded SnSe_2_ nanoparticles onto graphite nanosheets by high-energy ball milling, and they obtained SnSe_2_/graphite nanosheet nanocomposites. The SnSe_2_/graphite nanosheet nanocomposite exhibits an astonishing specific capacity (638.6 mA h g^−1^ after 100 cycles at 200 mA g^−1^) and rate capability (517.8 mA h g^−1^ at 5 A g^−1^) when used as the anode material for Na-ion batteries [101]. The ultra-fine SnSe_2_ nanoparticles and the elastic 3D carbon network enable fast ion transport, which results in high specific capacity and high multiplicity performance and structural stability.

### 2.3. Other Chalcogenides

#### 2.3.1. Other Sulfides

Other transition metal sulfides, such as manganese sulfide [102,103], titanium sulfide [104,105], bismuth sulfide [106,107,108] and multicomponent metal sulfides, are also potential anode materials, but they have not been widely reported.

There are three common crystal forms of MnS, namely α-MnS, β-MnS and γ-MnS. Zhu et al. successfully prepared a carbon-encapsulated core–shell structure of MnS@NSC composites as anode materials for sodium-ion batteries. The carbon shell provides a suitable space for the volume expansion of MnS during the reaction process, which ensures the structural stability of the material. Furthermore, the stable C-S-Mn bond makes an important contribution to the highly reversible reaction of the material. Therefore, the MnS@NSC composite achieves excellent cycling stability (329.1 mA h g^−1^ after at 1 A g^−1^ 3000 cycles) [102]. Liu et al. prepared α-MnS@N, S-NTC composites by the in situ encapsulation of α-MnS nanoparticles (NPs) in carbon nanotubes with N and S atoms for sodium-ion battery anode materials. The N, S-doped nanotubular carbon imparts high conductivity to the electrode material, and the unique one-dimensional bonding well mitigates the volume expansion of the electrode during the sodiation process. The α-Mns @ N, S-NTC exhibits a rate of 536 mA h^−1^ at a current density of 0.05 A g^−1^ during sodium-ion storage [103].

In order to solve the problem of mass change in the sodiation/desodiation process, multi-metal sulfide has been widely studied. Wang et al. synthesized fibrous Ni_3_S_2_@MOS_2_ composites by the hydrothermal method for sodium-ion battery anode materials. The special layered porous structure and the synergistic effect between the heterogeneous interfaces enable the Ni_3_S_2_@MoS_2_ anode material to exhibit a reversible capacity of 568 mA h g^−1^ at a current density of 200 mA g^−1^ [109]. Huang et al. designed a suitable MOF-engaged strategy and successfully synthesized a Bi_2_S_3_@CO_9_S_8_/NC composite with a unique core–shell for sodium ion battery anode material. The Bi_2_S_3_@Co_9_S_8_/NC composite spheres exhibit long-term cycling stability (458 mA h g^−1^ at 1 A g^−1^ after 1000 cycles) [110]. Cao et al. successfully prepared nitrogen-doped carbon matrix-covered dimetallic sulfide Sb_2_S_3_@FeS_2_ composites by a two-step solution method. When evaluated as anode materials for SIBs, the hollow and layered heterogeneous structure effectively enhanced Na^+^ diffusion and improved charge transfer at the heterogeneous interface. In addition, the synergistic coupling among Sb_2_S_3_, FeS_2_ and the external carbon layer shortens the electron/ion transport pathway and ensures structural stability during cycling. Therefore, Sb_2_S_3_@FeS_2_ as an anode material for SIBs exhibits 537.9 mA h g^−1^ rate capability at 10 A g^−1^ [111].

#### 2.3.2. Other Selenides

Bismuth selenide [112,113], copper selenide [114], and polymetallic selenide [115,116] are rarely studied for sodium ions. Lin et al. synthesized CuSe with crystalline columnar morphology (CPL-CuSe) by a solvothermal method and comprehensively investigated its sodium storage properties in SIBs. The structural transformation and phase change of CPL-CuSe as an anode material for sodium-ion batteries during the chemical reaction was systematically investigated by in situ XRD and in situ HRTEM. The reaction equation is as follows (Equation (56)):(56)$${\mathrm{CuSe}+2\mathrm{Na}}^{+}{\;+\;2e}^{-}\to {\mathrm{Cu}+\mathrm{Na}}_{2}\mathrm{Se}$$

Specifically, the CPL-CuSe delivers a specific capacity of 295 mA h g^−1^ at a current density of 10 A g^−1^ in SIBs [114].

In order to better improve the electrochemical performance of selenides, research shows that with the mixed-metal sulfide, different component synergy has better electrochemical properties than single-component metal sulfides. For instance, bimetallic heterostructure compounds show more abundant redox reactions and higher electronic conductivity as anode materials for sodium ion batteries. In general, heterostructures can introduce into the phase boundaries, which is the main reason for the increase in conductivity. The phase boundary is usually rich in lattice defects, distortions and dislocations, which have a significant impact on the electrochemical performance, and the phase boundaries can lower the activation barrier, so the reaction kinetics can be improved, thus enhancing electrochemical performance. Yang et al. designed VSe_2_/MoSe_2_ composites with heterostructures and obtained efficient sodium-ion storage [115]. Chen et al. fabricated hierarchical mesoporous MoSe_2_@CoSe/N-doped carbon (N-C) composites by in situ selenization of MoO_3_@Co-MOF precursors. The layered mesoporous structure enhances the Na^+^ diffusion rate and structural stability. The layered porous nanorod-structured composite exhibits excellent rate capability (485 mA h g^−1^ at 0.1 A g^−1^) and good specific capacity (398 mA h g^−1^ at 2 A g^−1^ after 1300 cycles) as an anode material for Na-ion batteries [117]. The detailed information of the electrochemical performance for various metal selenides in previous studies is shown in Table 1.

## 3. Summary and Outlook

In summary, this work reviews the research progress of MCs as advanced materials for high-performance SIBs in recent years. Compared with the previously reported literatures [21,118], detailed reaction mechanisms of various MCs during the sodiation/desodiation process have been summarized through advanced characterization techniques, such as in situ XRD, in situ TEM, ex situ XPS, etc. The results could be beneficial to providing significant guidance for fabricating high-performance MCs anodes for SIBs. Metal–sulfur compounds have higher theoretical capacity and weaker binding energy, lower reaction consumption, and they combine with sodium to produce Na_2_S and Na_2_Se with better electrical conductivity than Na_2_O. These indicate that metal–sulfur compounds offer great promise and advantages over other sodium-ion battery anodes. However, the large volume expansion associated with sodization and the conversion reaction that triggers a side reaction of the polysulfide with the electrolyte are problems that can lead to capacity degradation and safety issues. Therefore, carbon cladding, nanoengineering, heterostructuring, morphology controlling, dimensionality reduction, and electrolyte optimization are widely used to improve electrochemical performance. The carbon cladding of metal–sulfur compounds is considered to be the most commonly used strategy. Different carbon materials are chosen to provide stable frameworks and conductivity for metal–sulfide compounds. The nanostructural engineering of metal–sulfide compounds is an effective way to improve the capacity, reducing the particle size to shorten the diffusion length, and structural design at the micro- and nano-scale can improve the stability of the structure to mitigate bulk expansion. Transition metal-layered sulfur compounds have become very popular in recent years. The unique layered structures provide enough space for reversible sodium-ion sodization, resulting in superior electrochemical performance.

**Table 1 molecules-27-03989-t001:** The electrochemical performances of representative metal chalcogenides for SIBs.

Materials	Cycling Stability (mA h g^−1^)/ Current Density (A g^−1^)/ Cycles (*n*)	Rate Capability (mA h g^−1^)/ Current Density (A g^−1^)	Ref.
MoSe_2_@CoSe/N	347/2/300	392.8/2	[119]
FeS_2_@C yolk−shell	220/10/10,000	347/10	[120]
Yolk–shell N-doped carbon-coated FeS_2_	375/5/1000	307/10	[121]
Fe_7_S_8_@S/N–C	347/1/150	220/5	[122]
FeS_2_/NPCF	430/5/500	396/10	[123]
NHCFs/Fe_7_S_8_	517/2/1000	444/20	[124]
FeSe_2_/G	521.6/1/400	319.8/10	[125]
Rod-like FeSe_2_	308/10/10,000	308/10	[126]
Tailoring coral-like Fe_7_Se_8_@C	791.6/5/200	439.2/30	[127]
CoS_2_/C	610/0.5/120	391/2	[128]
NSPCF@CoS_2_	197/1/100	570/4	[129]
CoS_2_/CNTs/TiOxNy	72/1/100	166/2	[130]
CoS_2_/C polyhedrons	510//0.1/100	288/2	[131]
CoS_2_/C/C	652/1/500	572/5	[132]
CoS_2_@GC@B-CNT	432/5/900	419/10	[133]
rGO-CoS_2_-GNCSs	466/0.5/600	423/2	[134]
N–C/CoS_2_	698/1/500	458/10	[135]
CoSe_2_/CNFs	370/2/1000	224/15	[136]
CoSe_2_	414/0.2/700	362/5	[137]
CoSe_2_@3DSNC	409/5/1200	310/10	[138]
Cobblestone-like CoSe_2_@C nanospheres	345/4.5/10,000	355/2	[139]
CoSe_2_@N-CF/CNTs	428/1/500	406/10	[140]
CoSe_2_@C∩NC	234/5/2000	146/25	[141]
CNT/CoSe_2_/C	531/0.1/100	223.6/2.4	[142]
MoS_2_/graphite	244/0.1/800	225.4/1	[143]
Core–shell MoS_2_/C nanospheres	337/1/300	194/4	[144]
MoS_2_/graphene nanosheets	441/0.3/250	284/20	[145]
TiO_2_@C-MoS_2_	169/8/15,000	203/8	[146]
Ex-MoS_2_/RGO@C	415/0.1/150	316/2	[147]
C/N-MoS_2_-800	617.7/0.1/200	307/2	[148]
MoS_2_/GS-A	850/100/0.1	462/1	[149]
MoS_2_/NCF-MP	381/1/300	217/30	[150]
MoS_2_@graphene	309/10/2000	381/5	[151]
VM-43	451/2/800	207/20	[152]
BD-MoS_2_	350/2/1000	262/5	[153]
MoS_2_/rGO/S	190/2C/1000	350/2C	[154]
MoS_2_/G	421/0.3/250	284/20	[155]
Urchin-like MoS_2_/C	128/2/5000	413/5	[156]
NMF	407/1/100	432/1000	[157]
N-MoS_2_/C	401/0.2C/200	388/10C	[158]
Interlayer-expanded MoSe_2_ nanosheets	228/1/1500	333/2	[159]
MoSe_2_/rGO	247/0.1/100	214.1/1	[160]
MoSe_2_/N-PCD	437/0.2/500	266/2	[161]
MoSe_2_@C	384.2/5/2000	279/10	[162]
MoSe_2_@HBC	784.5/1/1200	298.8/20	[163]
MoSe_2_/P-C@TiO_2_	214/5/8000	202/10	[164]
MoSe_2_-G-CNTs	188/1/2000	241/2	[165]
MoSe_2_/N, P– C@N–C_3_MoSe_2_	238.7/5/5000	242/10	[166]
Hierarchical VS_2_ spheres	565/2/1000	533/4	[167]
DSV-VS_2−x_	175/5/5000	117/10	[168]
VS_4_/rGO	240/0.1/100	192/0.8	[169]
VS_4_/MWCNT	680/0.2/70	368/5	[170]
WSe_2_/CS	252/0.3/1200	122/1.5	[171]
WSe_2_/N, P-C-2	427/0.1/100	271/2	[172]
WSe_2_/C	257/0.1/90	114/2	[173]
NiS@C/rGO	240/0.1/500	216/2	[174]
NiS_2_	530/4/300	399/2	[175]
NiS@C	632/5/2000	688/3.2	[176]
NiS_2_NP/p-CNF	200/2/1000	300/2	[177]
NiS_x_@C-600	432/0.2/100	371/6.4	[178]
Ni_3_S_2_-NCF-0:1	378/1/100	286/5	[179]
NiS_2_@C@C.	580/0.1/100	448/1.6	[180]
Ni_3_S_2_/GO	249/0.1/50	115/5	[181]
Co-NiSe_2_/C	306/5/5000	260/10	[182]
SnS_2_-CM	322/0.2/200	257/1	[183]
MX/SnS_2_	322/0.1/200	134/1	[184]
SnS/C nanofibers	800/0.2/500	475/0.4	[185]
SnS_2_/CNT@rGO	301/1/1000	230/2	[186]
K-SnS_2_@SG	464/1/200	312/18	[187]
SSC@SnS_2_	564/0.1/100	411/5	[188]
SnS_2_/PCF	425/0.2/140	360/2	[189]
Sb_2_S_3_ added bio-carbon	220/1.0/200	208/2	[190]
Sb_2_Se_3_/C	378/0.4/50	270/0.8	[191]
ZnS/GAs	491/0.1/100	317/1	[192]
ZnS/N-CX	228/0.5/270	55/5	[193]
ZnS@d-PPy	148/1/100	203/5	[194]
N-ZnSe@rGO-L/N-ZnSe@rGO-H	285/0.3/500	338/5	[195]
ZnSe@C@rGO	223.7/5/1000	142/10	[196]
ZnSe/MWCNTs	246.7/1/600	280/10	[197]
CuS–CTAB	312/10/1000	172/20	[198]
CuS	517/5/2000	268/100	[199]
Cu_9_S_5_@NC	276/2/4000	237/5	[200]
Cu_2_S@NG	270/1.6/1000	260/3.3	[201]
Cu_2_S	317/5/5000	354/10	[202]
NiS_2_@CoS_2_	600/1/250	560/5	[203]
Bi_2_S_3_@Co_9_S_8_	354/0.5/500	293/5	[204]
SnO_2_@SnS_2_@NG	79/3/200	75/5	[205]
OL-VMS	260/1/130	453/5	[206]
FeS_2_/Fe_2_O_3_@N-CNF	287/1/600	246/3.2	[207]

However, the further development of sodium-ion batteries with metal–sulfur compounds requires more in-depth studies on the following aspects.

(1) Many metal–sulfide compound phase transitions are very complex and can have different effects on the electrochemical reactions. Some reaction mechanisms are still unclear, and there may be competing relationships, where different phase transitions within the first few cycles can cause different changes in the voltage plateau, affecting their electrochemical performance and hindering the commercialization of sodium-ion batteries. Therefore, a more in-depth mechanistic and kinetic study of metal–sulfur compounds is urgently needed.

(2) Most metal–sulfur compounds have a higher charging voltage plateau, which is safer than carbonaceous anode materials but also causes greater energy loss when used as anode materials for sodium-ion batteries. Therefore, high capacity and high energy density at relatively low voltages such as SnS_2_ can be better suited for the commercial development of sodium-ion batteries. It becomes necessary for batteries to find high-voltage cathode materials that match metal–sulfur compounds to achieve full cells with high-output energy density.

(3) The deeper reaction mechanisms of heterostructures in the (de)sodiation process still need to be further investigated; especially, the charge migration and electrochemical kinetics of these heterostructures are not yet fully understood and need to be studied in depth by advanced in situ/non-in situ characterization and theoretical calculations.

(4) The initial Coulomb efficiency reported in the text is more or less rarely above 85%, which makes the capacity decay too fast and the capacity too low. Therefore, a protective layer is needed to form a stable SEI layer and thus improve the electrochemical performance of the metal–sulfide compounds. The electrolyte plays a key role in forming a stable SEI layer, and most of the previous literature is on ester-based electrolytes; however, the electrochemical effects of ether-based electrolytes and highly concentrated electrolytes on sodium ion batteries have not been reported much and need to be further investigated.

(5) In order to make the synthesis route simple for commercial application, on the one hand, it is necessary to use some non-toxic, high-resource and low-cost metal–sulfur compounds. Therefore, materials with high abundance and low cost such as FeS_2_ and Ni_3_S_2_ are more worthy of research and are more conducive to commercialization. On the other hand, the synthesis routes of metal–sulfur compounds are more complex, with long reaction times and demanding experimental conditions, so some simple and scalable synthesis routes are more favorable for commercial applications, such as ball milling, spray drying and spray pyrolysis, which are effective methods for the large-scale production of metal–sulfur compound anode materials with low cost and simple and scalable synthesis routes.

(6) In real-world applications, sodium-ion batteries should operate in the temperature range of −20 to 60 °C and below 0 °C, with few reports of tests above room temperature, and below 0 °C, the viscosity of the adhesive drops sharply, and the sodium ions move slowly. In contrast, high temperature causes enhanced side reactions between the negative electrode material and the electrolyte. Therefore, more attention should be paid to low and high temperatures in the future research on metal–sulfur compounds.

(7) Theoretical calculations and theoretical simulations are essential to predict the thermodynamics and kinetics of metal–sulfide compounds during the reaction of sodium ion batteries. These theoretical studies include MD and DFT calculations, and once their results are consistent and can well confirm the experimental results, they can provide new ideas for battery development and thus optimize the design of new electrode materials. Advanced characterization techniques become more important for the electrochemical mechanisms and failure mechanisms of metal–sulfur compounds. The mechanisms of sodization and denaturation of different metal–sulfur compounds are still unclear. Therefore, various in situ and non-in situ characterization techniques are needed, which can play a key role in elucidating the structural evolution of materials and the corresponding electrochemical processes.

In conclusion, although metal–sulfide composites still face great challenges in practical applications, it is reasonable to expect that better fundamental understanding and more advanced detection techniques will pave the way for commercial applications of high-performance sodium-ion batteries.

## Figures and Tables

**Figure 2 molecules-27-03989-f002:**
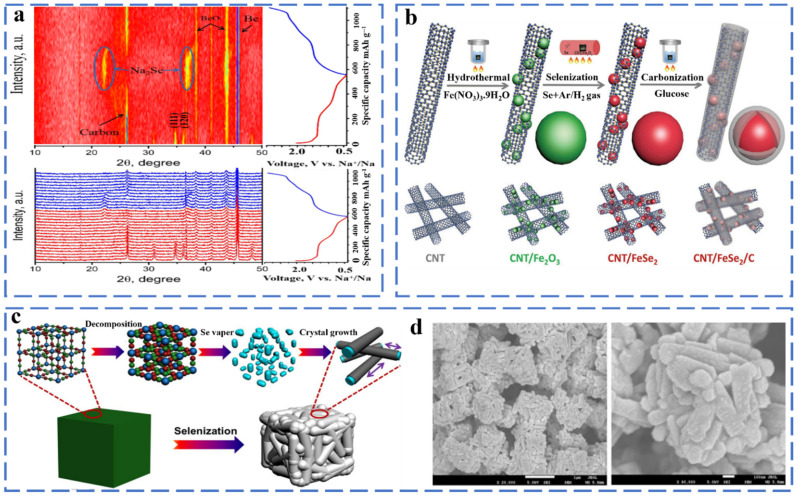
(**a**) In situ XRD results contour plot and corresponding in situ XRD patterns and discharge/charge curves of FeSe_2_@NC MR electrode during initial cycling. Reprinted with permission from Ref. [34]. Copyright 2020, copyright Pan, Q. (**b**) Preparation process of CNT/FeSe_2_/C. Reprinted with permission from Ref. [35]. Copyright 2020, copyright Yousaf, M. (**c**) The formation mechanism of 3D hierarchical iron selenide hollow nanocubes assembled from FeSe_2_@C core–shell nanorods, (**d**) SEM images. Reprinted with permission from Ref. [36]. Copyright 2018, copyright Fan, H.

**Figure 3 molecules-27-03989-f003:**
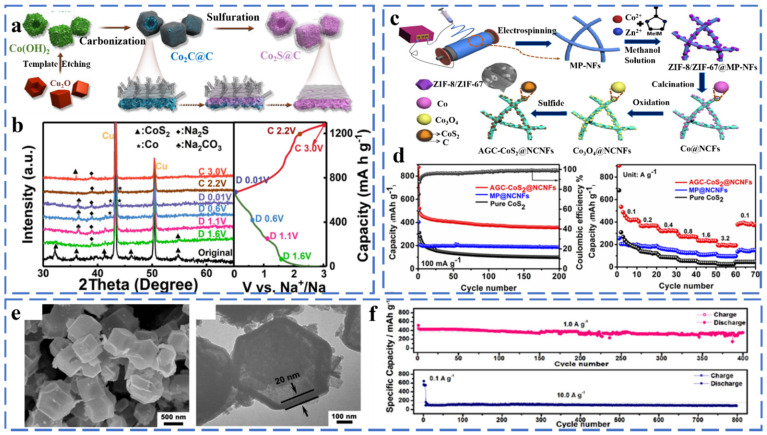
(**a**) Schematic illustration of the synthesis route for the CoS_2_@C composite, (**b**) Ex situ XRD patterns of the CoS_2_@C at various discharging/charging states. Reprinted with permission from Ref. [39]. Copyright 2020, copyright Zhao, Z. (**c**) Schematic illustration of the synthesis of AGC-CoS_2_@NCNFs, (**d**) Cycling performance and rate capability of AGC-CoS_2_@NCNFs, pure CoS_2_ and MP@NCNFs. Reprinted with permission from Ref. [40]. Copyright 2020, copyright Zhang, W. (**e**) SEM and TEM images (**f**) long-term cycle stability at 1.0 and 10.0 A g^−1^ of Co_9_S_8_@CYSNs. Reprinted with permission from Ref. [41]. Copyright 2020, copyright Zhao, Y.

**Figure 4 molecules-27-03989-f004:**
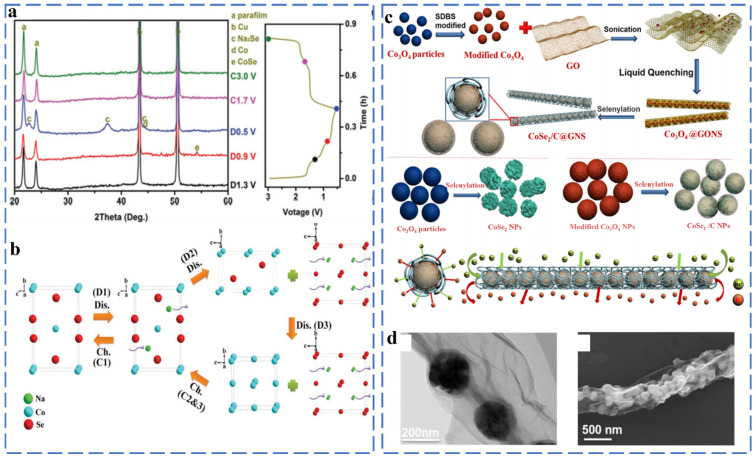
(**a**) Ex situ XRD patterns of the CoSe_2_ at various discharging/charging states. (**b**) Schematic illustration of reaction mechanism of CoSe_2_ when cycling between 0.5 and 3.0 V and using ether-based electrolytes. Reprinted with permission from Ref. [43]. Copyright 2016, copyright Zhang, K. (**c**) Schematic illustration of the fabrication of CoSe_2_/C@GNS, (**d**) SEM and TEM images. Reprinted with permission from Ref. [44]. Copyright 2016, copyright Zhang, Z.

**Figure 5 molecules-27-03989-f005:**
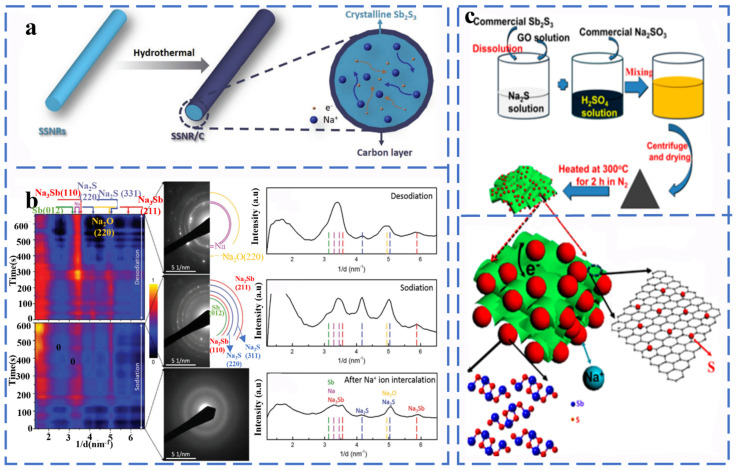
(**a**) Schematic representation of the synthesis process of SSNRs and SSNR/C nanocomposite, (**b**) Phase evolution during the first sodiation and desodiation processes of SSNR/C electrode probed by in situ electron diffraction. Reprinted with permission from Ref. [51]. Copyright 2020, copyright Wang, P. (**c**) Schematic illustration of the fabrication process of the Sb_2_S_3_/SGS composite. Reprinted with permission from Ref. [52]. Copyright 2018, copyright Zhao, Z.

**Figure 6 molecules-27-03989-f006:**
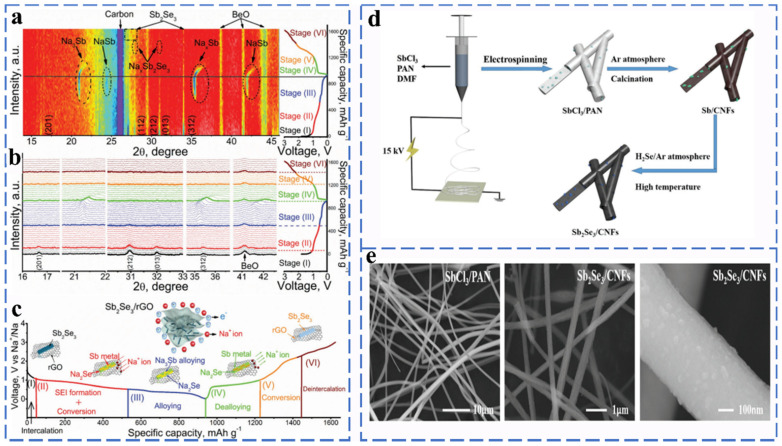
(**a**) In situ XRD patterns of Sb_2_Se_3_/rGO anode operated at different states of charge against the voltage during the initial cycle. (**b**) XRD patterns within a selected 2θ range during the first cycle against the voltage. (**c**) Schematic of the reaction mechanism of the Sb_2_Se_3_/rGO hybrid during the charge/discharge process. Reprinted with permission from Ref. [53]. Copyright 2019, copyright Hou, M. (**d**) Schematic illustration of the fabrication process of Sb_2_Se_3_/CNFs. (**e**) SEM image. Reprinted with permission from Ref. [54]. Copyright 2021 copyright Luo, F.

**Figure 7 molecules-27-03989-f007:**
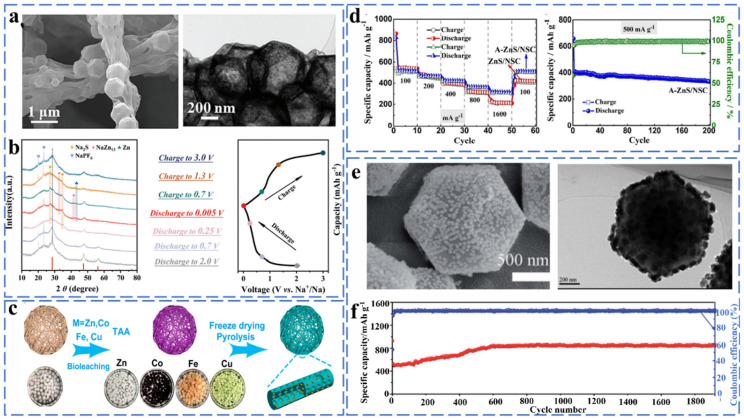
(**a**) SEM images at different resolutions and a TEM image of the 1.5-ZnS@CNFs-800 composite. (**b**) Ex situ XRD patterns at different cut-off voltages. Reprinted with permission from Ref. [56]. Copyright 2019, copyright. Wang, J. (**c**) Schematic illustration of the formation of the MS/NCF (MS = ZnS, Co_9_S_8_, FeS, Cu_1.81_S). Reprinted with permission from Ref. [57]. Copyright 2021 copyright. Yuan, Z. (**d**) Rate performances of ZnS/NSC and A-ZnS/NSC samples at different current densities of 100, 200, 400, 800 and 1600 mA g^−1^, respectively, and cycling performance and Coulombic efficiency of A-ZnS/NSC at 500 mA g^−1^. Reprinted with permission from Ref. [58]. Copyright 2021 copyright. Li, Z. (**e**) SEM and TEM image of ZnS/NPC. (**f**) Cycling performance of ZnS/NPC and ZnS at 0.1 A g^−1^. Reprinted with permission from Ref. [59]. Copyright 2019 copyright. Wang, S.

**Figure 8 molecules-27-03989-f008:**
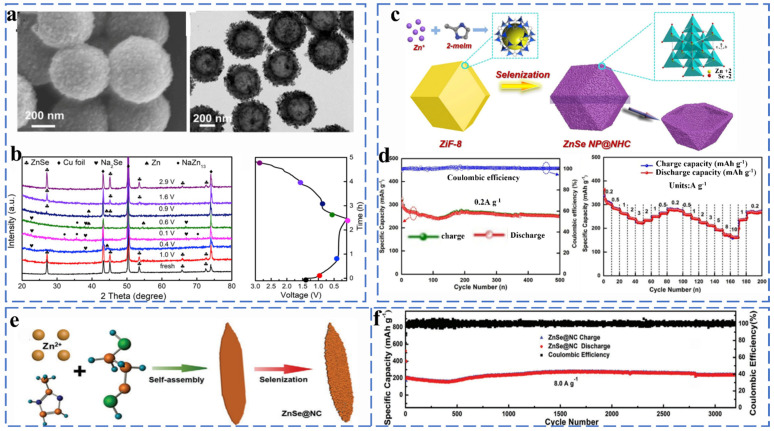
(**a**) SEM and TEM image of ZnSe@HCNs. (**b**) Ex situ XRD analysis of the ZnSe@HCNs electrodes at different charge/discharge states and corresponding charge/discharge curve. Reprinted with permission from Ref. [60]. Copyright 2019 copyright. Zheng, F. (**c**) Schematic illustration for the fabrication processes and structure of ZnSe NP@NHC nanocomposite. (**d**) Cycling performances and the corresponding Coulombic efficiencies and rate performance. Reprinted with permission from Ref. [61]. Copyright 2019 copyright. Yousaf, M. (**e**) Schematic illustration of the formation process of ZnSe@NC. (**f**) The ultralong cycling performances of ZnSe@NC at 8.0 A g^−1^. Reprinted with permission from Ref. [62]. Copyright 2018 copyright. Liu, H.

**Figure 9 molecules-27-03989-f009:**
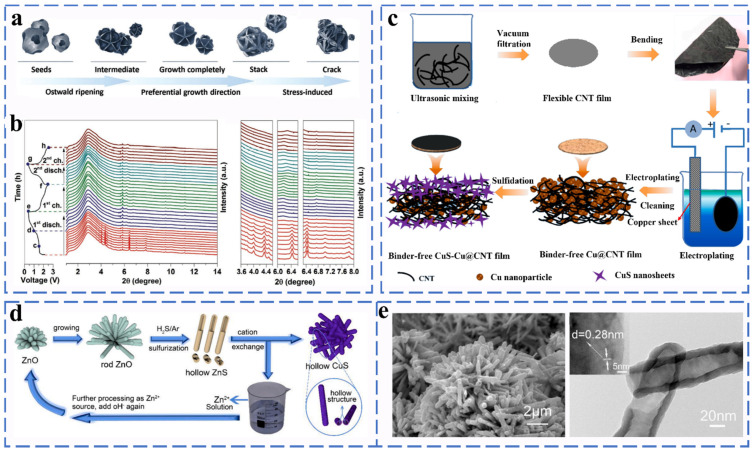
(**a**) Schematic illustration of the formation mechanism of CuS spheres. (**b**) In situ SRD patterns of CuS electrode collected at first and second cycle charge–discharge processes between 0.4 and 2.6 V. Diffraction peak changes at various 2 theta degrees at first and second cycle charge–discharge states. Reprinted with permission from Ref. [63]. Copyright 2018 copyright. Ge, P. (**c**) Schematic of the fabrication process for the CuS-Cu@CNTs membrane. Reprinted with permission from Ref. [64]. Copyright 2018 copyright. Jeong. S. (**d**) SEM image of zinc oxide prepared by reacting at 60 °C for 4 h. (**e**) SEM and TEM image for the preparation of zinc oxide at 60 °C for 8 h. Reprinted with permission from Ref. [65]. Copyright 2018 copyright. Zhao, X.

**Figure 10 molecules-27-03989-f010:**
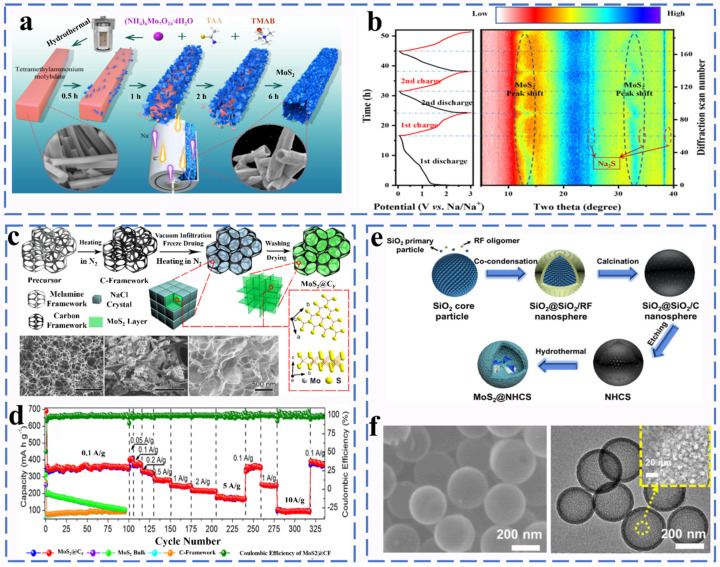
(**a**) Schematic illustration of the tubular MoS_2_ hierarchical structures. (**b**) In situ XRD patterns for MS-T upon sodiation and desodiation. Reprinted with permission from [66]. Copyright 2020 copyright. Zhao, W. (**c**) Schematic illustrations of the synthetic process of MoS_2_@CF samples and (**d**) SEM images. Reprinted with permission from [67]. Copyright 2018 copyright. Yu, D. (**e**) Mechanism for the fabrication process of MoS_2_@NHCS and (**f**) SEM images. Reprinted with permission from [68]. Copyright 2017 copyright. Sun, R.

**Figure 11 molecules-27-03989-f011:**
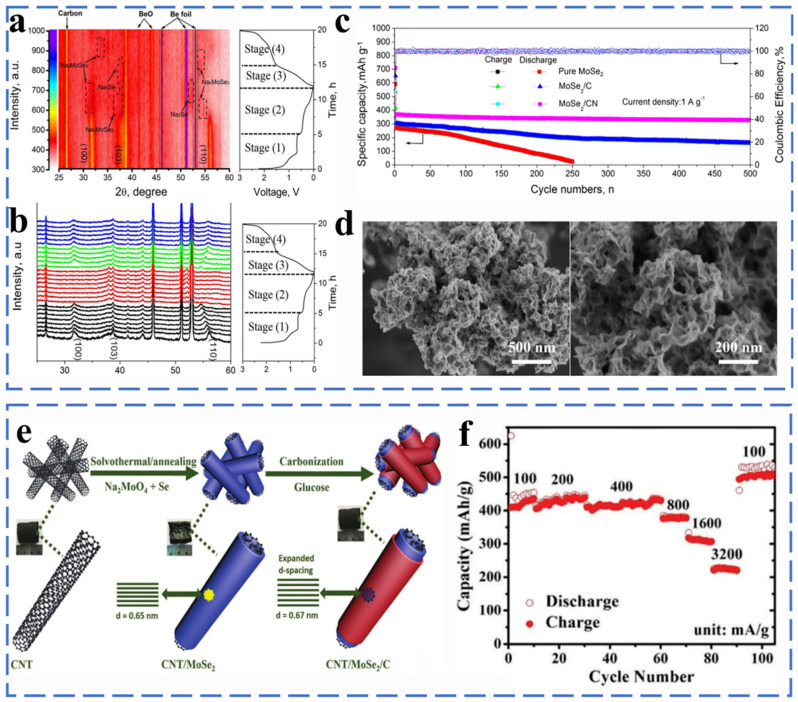
(**a**) Contour plots of in situ XRD results of the MoSe_2_ electrode against the voltage profile during the initial cycle at cut-off voltage of 0.01–3.0 V. (**b**) Corresponding in situ XRD patterns of the MoSe_2_. (**c**) CV curves of the MoSe_2_/CN electrode before cycling and after 500th between 0.01 and 3.0 V at a scanning rate of 0.1 mV s^−1^. (**d**) SEM images. Reprinted with permission from [75]. Copyright 2020 copyright. Kim, I. (**e**) Schematic representation for the synthesis process of 3D trilayer framework design from CNT framework to CNT/MoSe_2_/C. (**f**) Rate capability of CNT@MoSe_2_/C. Reprinted with permission from [76]. Copyright 2018 copyright Cho, J.

**Figure 12 molecules-27-03989-f012:**
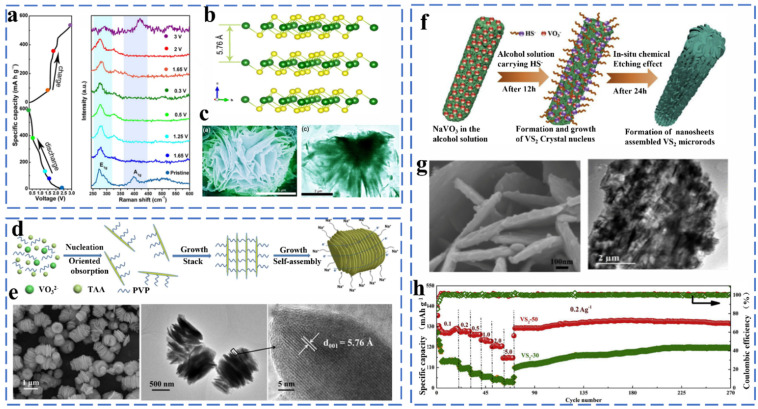
(**a**) The discharge–charge curves of the first cycle and ex situ Raman spectrum at different discharge–charge states, (**b**) Schematic crystal structure of VS_2_. (**c**) SEM and TEM image. Reprinted with permission from [82]. Copyright 2017 copyright Zhou, P. (**d**) Schematic illustration of the formation of layer-by-layer VS_2_-SNSs. (**e**) SEM image and TEM image and HRTEM image of VS_2_-SNSs. Reprinted with permission from [83]. Copyright 2018 copyright Cui, J. (**f**) Schematic illustration of the in situ chemical etching process of nanosheets assembled 3D hierarchical VS_2_ microrods. (**g**) SEM and TEM image. (**h**) Rate and cycling performances. Reprinted with permission from [84]. Copyright 2018 copyright Xue, P.

**Figure 13 molecules-27-03989-f013:**
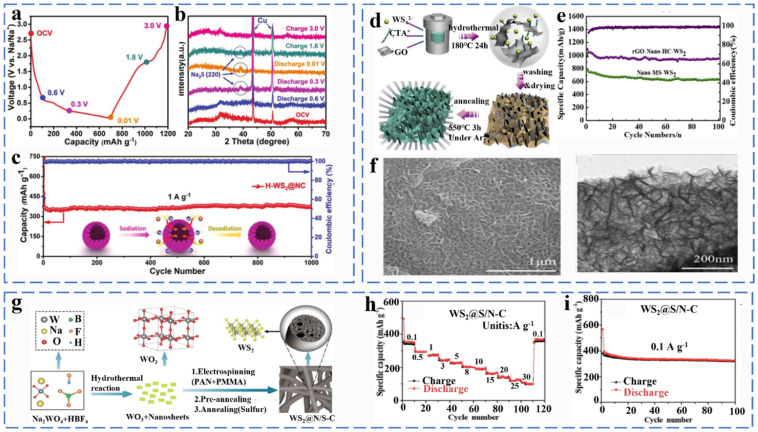
(**a**) Charging/discharging profiles of initial cycle at 0.1 A g^−1^. (**b**) The ex situ XRD patterns of H-WS_2_@NC electrode at different cut-off potentials during the initial discharge/charge process. (**c**) Long-term cycling performance at a current density of 1.0 A g^−1^ for the H-WS_2_@NC electrode. Reprinted with permission from [87]. Copyright 2017 copyright Yao, S. (**d**) Synthesis process of WS_2_/rGO Nano-HC. (**e**) Cycling performances of the WS_2_ Nano-MS and WS_2_/rGO Nano-HC at low current density of 100 mAg^−1^. (**f**) SEM and TEM image. Reprinted with permission from [88]. Copyright 2016 copyright Xiong, X. (**g**) Schematic illustration of the synthetic route to the porous WS_2_@S/N–C nanofibers. (**h**) Rate capability of the WS_2_@S/N–C electrodes at various current densities. (**i**) WS_2_@S/N–C electrodes at a current density of 0.1 A g^−1^. Reprinted with permission from [89]. Copyright 2019 copyright Yao, S.

**Figure 14 molecules-27-03989-f014:**
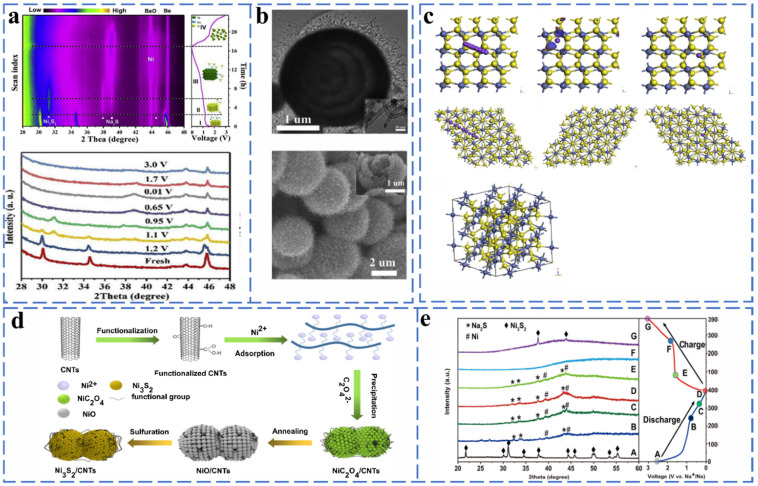
(**a**) In situ XRD patterns during the first charge and discharge process and XRD patterns at different discharge–charge stages of the NiS@NCNT MSHMs electrodes. (**b**)TEM and SEM image. Reprinted with permission from [92]. Copyright 2018 copyright Fang, Y. (**c**) Geometric models for Na ion migration pathways on the NiS_2_ (100) surface and NiS_2_ (111) surface of top views and optimized geometry structures of NiS_2_ and corresponding diffusion energy barrier profiles on different surfaces. Reprinted with permission from [93]. Copyright 2021 copyright Wei, X. (**d**) The synthetic procedure of Ni_3_S_2_/CNTs micro-nanostructure composites. (**e**) Ex situ XRD curves of Ni_3_S_2_/CNTs-10 electrode at different states in the first cycle. Reprinted with permission from [94]. Copyright 2019 copyright Li, J.

**Figure 15 molecules-27-03989-f015:**
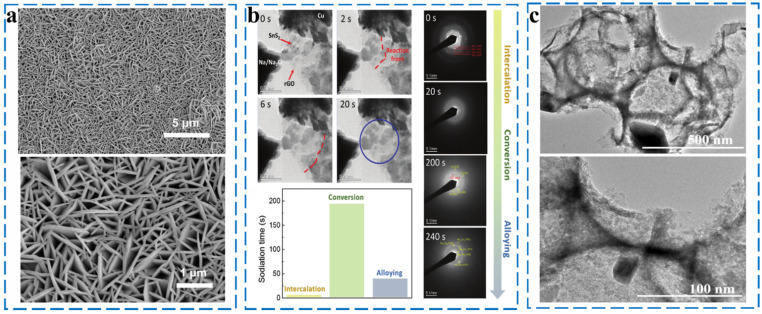
(**a**) SEM image of SnS_2_ NWAs. Reprinted with permission from [97]. Copyright 2019 copyright Lu, S. (**b**) Sequential TEM images of Na intercalation into the SnS_2_/GCA electrode taken from the high-speed video and SAED patterns taken at different sodiation stages and durations required to complete intercalation, conversion, and alloying reactions during the in situ sodiation process. Reprinted with permission from [98]. Copyright 2019 copyright He, Y. (**c**) Low and high-resolution TEM images of the 1.5–350 3D SnS/C composite. Reprinted with permission from [99]. Copyright 2021 copyright Sun, Y.

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
