# Peer review of "Innovative Materials for Energy Storage and Conversion"

_molecules, 2022, doi:10.3390/molecules27133989_

Round 1

Reviewer 1 Report

Title: Innovative Materials for Energy Storage and Conversion

Authors: Shi Li, Shi Luo, Liya Rong, Linqing Wang, Ziyang Xi, Yong Liu, Yuheng Zhou, Zhongmin Wan and Xiangzhong Kong

Summary:

Metal chalcogenides for sodium-ion batteries are reviewed in detail, while the challenges and possible solutions are summarized in the end of the manuscript. The review is well written with only a few language errors that should be corrected. I have a few questions/suggestions as follows: 

Issues:

  1. (page 1, rows 42-43) ‘To date, metal chalcogenides have been extensively studied and metal chalcogenides include metal sulfides and metal selenides.’ This phrase is not clear and should be modified. Metal chalcogenides also include metal tellurides.
  2. (page 2, row 80) Change ‘VanderWaals’ to van der Waals.
  3. (page 2, row 76-91) Authors discuss two types of chalcogenide materials, layered and non-layered in this order, and then they discuss both of them in opposite order. They should be consistent.
  4. (pages 2, 4 rows 92, 142) Reports show that FeS2 and FeSe2 are layered chalcogenide materials [https://doi.org/10.1515/nanoph-2020-0014; https://doi.org/10.1002/adma.202008456]. Why are these materials included in the Non-layer structured MCs?
  5. (page 30, row 964) Table 1 is not cited in the text of the manuscript.
  6. As a general comment, the figures do not have the necessary resolution. Quality should be improved, some of them are not even visible and magnification is not possible.

Author Response

We would like to thank the reviewer’ valuable comments which would definitely help us improve the quality of our manuscript. We have studied the comments carefully and revised the manuscript accordingly. The detailed changes are highlighted in yellow background in revised manuscript. The point-to-point responses are listed as follows:

Reviewer:

Metal chalcogenides for sodium-ion batteries are reviewed in detail, while the challenges and possible solutions are summarized in the end of the manuscript. The review is well written with only a few language errors that should be corrected. I have a few questions/suggestions as follows:

Answer: Thank the reviewer for the comment.

  1. (page 1, rows 42-43) ‘To date, metal chalcogenides have been extensively studied and metal chalcogenides include metal sulfides and metal selenides.’ This phrase is not clear and should be modified. Metal chalcogenides also include metal tellurides.

Answer: Thank the reviewer for the comment. "To date, metal chalcogenides have been extensively studied and metal chalcogenides include metal sulfides and metal selenides." This statement has been revised in the text to "To date, metal sulfides and metal selenides have been extensively studied."

  1. (page 2, row 80) Change ‘VanderWaals’ to van der Waals.

Answer: Thank the reviewer for the comment. The 'VanderWaals' has been changed as 'van der Waals' in the text.

  1. (page 2, row 76-91) Authors discuss two types of chalcogenide materials, layered and non-layered in this order, and then they discuss both of them in opposite order. They should be consistent.

Answer: Thank the reviewer for the comment. The content of lines 76-89 in the text has been changed as “Metal chalcogenides can generally be divided into two parts: non-layered material and layered structure material. The non-layered substances are also very popular due to its low price and high theoretical capacity advantages. non-layered substances crystallize in three dimensions through atoms or chemical bonds, reflecting the non-layered nature of their bulk crystals. Layer-structured substances have been focused more attention due to the unique layered structure and excellent electrochemical properties. For layered materials, the strong chemical bonds in each layer connect the in-plane atoms to each other, and these layers are stacked together through weak Van der Waals force interactions to form bulk crystals. Besides, weaker interaction be-tween the guest ions and chalcogenide lattice for layered structures enables faster ion migration, lowering the energy barrier required to initiate the intercalation reaction. At present, numerous methods, such as solvothermal reaction, spray pyrolysis, chemical vapor deposition (CVD), electrospinning, exfoliation, sul-fation/selenization, reflux, ball milling et al. have been utilized for the synthesis of nanostructured MCs materials.”

  1. (pages 2, 4 rows 92, 142) Reports show that FeS2 and FeSe2 are layered chalcogenide materials [https://doi.org/10.1515/nanoph-2020-0014; https://doi.org/10.1002/adma.202008456]. Why are these materials included in the Non-layer structured MCs?

Answer: Thank the reviewer for the comment. By consulting a large number of literatures, it is found that whether FeS2 and FeSe2 are layered structures is controversial. For example, Wu et al. listed FeS2 and FeSe2 as non-layered structures.       [doi: https://doi.org/10.1016/j.ensm.2020.10.007].

  1. (page 30, row 964) Table 1 is not cited in the text of the manuscript.

Answer: Thank the reviewer for the comment. Table 1 is summarized in row 877-878.

“The detailed information of the electrochemical performance for various metal selenides in previous studies has been shown in Table 1.”

  1. As a general comment, the figures do not have the necessary resolution. Quality should be improved, some of them are not even visible and magnification is not possible.

Answer: Thank a lot for the reviewer’s suggestions. Figure 2,3,4,5,6,9,11,12,13,15 have been modified in the manuscript as follows:

The detailed informations about point-by-point response please see the attchment.

Reviewer 2 Report

The authors in this manuscript has been reviewed the metal chalcogenides sodium batteries. The last five years have been appeared several reviews and book chapters related to this subject. This review do not add any additional information to the  existing. The manuscript is written carelessly with a lot of mistakes. The figures are of low resolution the letters on the figures cannot be read. The table is complete wrong with the values to be mixed. The conclusions are ok however, I do not think we need  currently another review on this subject.

Author Response

We would like to thank the reviewer’ valuable comments which would definitely help us improve the quality of our manuscript. We have studied the comments carefully and revised the manuscript accordingly. The detailed changes are highlighted in yellow background in revised manuscript. The point-to-point responses are listed as follows:

Reviewer:

The authors in this manuscript has been reviewed the metal chalcogenides sodium batteries. The last five years have been appeared several reviews and book chapters related to this subject. This review do not add any additional information to the existing. The manuscript is written carelessly with a lot of mistakes. The figures are of low resolution the letters on the figures cannot be read. The table is complete wrong with the values to be mixed. The conclusions are ok however, I do not think we need currently another review on this subject.

Answer: Thank the reviewer for the comment. We apologized for the numerous spell and grammar mistakes and fuzzy figures in the manuscript. After carefully checking, the mistakes have been corrected and all the figures have been adjusted to make them clear.

Recently, metal chalcogenides with advantages of low cost and high theoretical capacity have been considered to be one of the most popular anode materials for advanced sodium ion batteries. However, the poor electrochemical stability and slow kinetic behaviors hinder their practical applications. Several previous reviews have summarized the strategies for improving the sodium storage properties of the metal chalcogenides. Nevertheless, detailed the reaction mechanisms of various metal chalcogenides during sodiation/de-sodiation process have rarely been analyzed and summarized. Besides, advanced characterization techniques such as in situ XRD, in situ TEM et al. can detect the electrochemical reaction mechanism and structural characterizations during charge/discharge process of the electrodes, which has clear theoretical guidance for developing high performance metal chalcogenides anodes for sodium-ion batteries. In our review, profound and detailed clarifications and summaries of the reaction mechanisms of anode materials have been reported and advanced characterization techniques which has clear theoretical guidance for metal chalcogenides anodes have been introduced. The manuscript has been revised as follows:

The detailed revisions about figures, text, references and table can be seen in the attachment.

Reviewer 3 Report

Could be accepted with minor review.

1.       Check reference: REF 34, 69, 71, 204. This reference should be cited and discussed: Chemistry – A European Journal 22 (2016) 590-597.

2.       Figures are in low quality, could not read. Please try to improve the resolution of all figures. The abbreviation should be explained in the caption of Figure 5.

3.       Careful proof reading is needed.

4.       A comparison of this review with similar reviews should be clearly stated.

5.       It is suggested to add an eye-catching figure 1 to summarize the review.

Author Response

We would like to thank the reviewer’ valuable comments which would definitely help us improve the quality of our manuscript. We have studied the comments carefully and revised the manuscript accordingly. The detailed changes are highlighted in yellow background in revised manuscript. The point-to-point responses are listed as follows:

Reviewer:

  1. Check reference: REF 34, 69, 71, 204. This reference should be cited and discussed: Chemistry – A European Journal 22 (2016) 590-597.

Answer: Thank the reviewer for the comment. References: REF 34, 69, 71, 204 have been checked and revised. The corresponding reference "Chemistry - A European Journal 22 (2016) 590-597" has been cited and discussed.

[34]. Pan, Q.; Zhang, M.; Zhang, L.; Li, Y.; Li, Y.; Tan, C.; Zheng, F., Huang, Y.; Wang, H.; Li, Q., FeSe2@C microrods as a superior long-life and high-rate anode for sodium ion batteries. ACS Nano 2020, 14, 17683–17692.

[69]. Li, W.; Huang, J.; Feng, L.; Cao, L.; Liu, Y.; Pan, L., Nano-grain dependent 3D hierarchical VS2 microrods with enhanced intercalation kinetic for sodium storage properties.  J. Power Sources 2018, 398, 91-98.

[71].      Wu, Y.; Zhong, W.; Tang, W.; Zhang, L.; Chen, H.; Li, Q.; Xu, M.; Bao, S., Flexible electrode constructed by encapsulating ultrafine VSe2 in carbon fiber for quasi-solid-state sodium ion batteries. J. Power Sources 2020, 470.228438.

[204]. Yue, X.; Wang, J.; Xie, Z.; He, Y.; Liu, Z.; Liu, C.; Hao, X.; Abudula, A.; Guan, G., Controllable synthesis of novel orderly layered VMoS2 anode materials with super electrochemical performance for sodium-ion batteries. ACS appl. Mater. Inter. 2021, 13, 26046-26054.

[108]. Li, W., Han, C., Chou, S., Wang, J., Li, Z., Kang, Y., Dou, S. Graphite-nanoplate-coated Bi2S3 composite with high-volume energy density and excellent cycle life for room-temperature sodium-sulfide batteries. Chem.-Eur. J., 22, 590–597.

  1. Figures are in low quality, could not read. Please try to improve the resolution of all figures. The abbreviation should be explained in the caption of Figure 5.

Answer: Thank the reviewer for the comment. All graphics in the text have been replaced by high-resolution graphics to achieve a good visual effect.

The noun abbreviations in Figure 5 have been explained in the manuscript.

Xiong et al. synthesized Sb2S3/SGS composites by rivets of nanostructured Sb2S3 on sulfur-doped graphene sheets(SGS) (Figure 5c).

  1. Careful proof reading is needed.

Answer: Thank the reviewer for the comment. We have carefully checked and proofread the manuscript. The spell and grammar mistakes have been corrected.

  1. A comparison of this review with similar reviews should be clearly stated.

Answer: Thank the reviewer for the comment. We have made changes in the manuscript, which are as follows.

In summary, this work reviews the research progress of metal-sulfur compounds as motor materials for high-performance sodium ion batteries in recent years. Compared with the previously reported literatures[21,118], we have carried out a more profound and detailed clarification and summary of the reaction mechanism of anode materials through advanced characterization techniques, which has clear theoretical guiding significance for chalcogenides as anode materials for sodium-ion batteries.

[118].     Fang, Y., D. Luan, and X.W.D. Lou, Recent advances on mixed metal sulfides for advanced sodium-ion batteries. Adv Mater, 2020. 32: e2002976.

  1. It is suggested to add an eye-catching figure 1 to summarize the review.

Answer: Thank the reviewer for the comment. Summary graph has been added as follows.

The detailed revisions about figures can be seen in the attachment.

Round 2

Reviewer 2 Report

The authors have revised the manuscript to a much more improved. 

However a few points to considered.

1) The importance  and the differences of this review from previous reviews on the same subject should addressed in the introduction not to me. Same thing in the conclusions, what new this review have been added in literature and how this is related to the future

2) Table 1 still has several mistakes. The number of cycles, current etc are in wrong positions

3) The figures still are difficult to be read. Change the way are presented 

Author Response

We would like to thank the reviewer’ valuable comments which would definitely help us improve the quality of our manuscript. We have studied the comments carefully and revised the manuscript accordingly. The detailed changes are highlighted in yellow background in revised manuscript. The point-to-point responses are listed as follows:

Reviewer:

The authors have revised the manuscript to a much more improved. However, a few points to considered.

1) The importance and the differences of this review from previous reviews on the same subject should addressed in the introduction not to me. Same thing in the conclusions, what new this review has been added in literature and how this is related to the future.

Answer: Thank the reviewer’ suggestions. We have made the corresponding changes in the introduction and conclusions. The manuscript has been changed as follows:

  1. Introduction:

This work reviews the research progress of MCs as advanced materials for high-performance SIBs in recent years. The synthesis method, structure composition, reaction mechanism, in-situ characterization and performance modification methods are discussed. Moreover, based on the structure and conductivity of metal chalcogenides, we summarized several strategies to suppress the drawbacks of MCs as anode materials for SIBs, such as designing nanostructure with various morphologies, carbon coating, hierarchical structure and hollow porous structure. Finally, a brief discussion on the future development and perspectives of metal chalcogenides materials as advanced anodes for SIBs has been made. Compared with the previously reported literatures, we have elucidated and summarized the detailed reaction mechanisms of numerous MCs during sodiation/desodiation process based on advanced in-situ/ex-situ characterization techniques, which provides significant theoretical guidance for developing MCs with enhanced sodium storage properties.

Summary and Outlook: In summary, this work reviews the research progress of MCs as advanced materials for high-performance SIBs in recent years. Compared with the previously reported literatures[21, 118], detailed reaction mechanisms of various MCs during sodiation/desodiation process have been summarized through advanced characterization techniques, such as in situ XRD, in situ TEM, ex situ XPS, et al. The results could be beneficial to providing significant guidance for fabricating high performance MCs anodes for SIBs.

2) Table 1 still has several mistakes. The number of cycles, current etc are in wrong positions

Answer: Thank the reviewer for the comment. We apologize for the errors in the table. The Table 1 has been adjusted and the manuscript has been revised as follows:

Table 1. The electrochemical performances of representative metal chalcogenides for SIBs. (seen in the attachment)

3) The figures still are difficult to be read. Change the way are presented.

Answer: Thank the reviewer for the comment. We have corrected the unclear images and the manuscript has been revised.

The detailed revised information can be seen in the attachment.
